



# Tree-roots control of shallow landslides

Denis Cohen[1] and Massimiliano Schwarz[2]

[1]Department of Earth and Environmental Science, New Mexico Tech, Socorro, NM 87801, USA
[2]School of Agricultural, Forest, and Food Sciences, Bern University of Applied Science, 3052, Zollikofen, Switzerland

*Correspondence to:* Denis Cohen (denis.cohen@gmail.com)

**Abstract.** Tree roots have long been recognized to increase slope stability by reinforcing the strength of soils. Slope stability models include the effects of roots by adding an apparent cohesion to the soil to simulate root strength. No model includes the combined effects of root distribution heterogeneity, stress-strain behavior of root reinforcement, or root strength in compression. Recent field observations, however, indicate that shallow landslide triggering mechanisms are characterized by differential
deformation that indicates localized activation of zones in tension, compression, and shear in the soil. These observations contradict the common assumptions used in present models. Here we describe a new model for slope stability that specifically considers these effects. The model is a strain-step discrete element model that reproduces the self-organized redistribution of forces on a slope during rainfall-triggered shallow landslides. We use a conceptual sigmoidal-shaped hillslope with a clearing in its center to explore the effects of tree size, spacing, weak zones, maximum root-size diameter, and different root strength
configurations. The model is driven by root data of Norway spruce obtained from laboratory and field measurements. Simulation results indicate that tree roots can stabilize slopes that would otherwise fail without them and, in general, higher root density with higher root reinforcement results in a more stable slope. Root tension provides more resistance to failure than root compression but roots with both tension and compression offer the best resistance to failure. Lateral (slope-parallel) tension can be important in cases when the magnitude of these forces is comparable to the slope-perpendicular tensile forces. In these
cases, lateral forces can bring to failure tree-covered areas with high root reinforcement. Slope failure occurs when downslope soil compression reaches the soil maximum strength. When this occurs depends on the amount of root tension upslope in both the slope-perpendicular and slope-parallel directions. Roots in tension can prevent failure by reducing soil compressive forces downslope. When root reinforcement is limited, hillslopes form a crack parallel to the slope near its top. Simulations with roots that fail across this crack always resulted in a landslide. Slopes that did not form a crack could either fail or remain stable,
depending on root reinforcement. Tree spacing is important for the location of weak zones but tree location on the slope (with respect to where a crack opens) is as important. Finally, for the specific cases tested here, large roots, greater than 20 mm, are too few too contribute significantly to root reinforcement. Omitting roots larger than 8 mm predicted a landslide when none should have occurred. Intermediate roots (5 to 20 mm) appear to contribute most to root reinforcement and should be included in calculations.

To fully understand the mechanisms of shallow landslide triggering requires a complete re-evaluation of the traditional apparent-cohesion approach that does not reproduce the incremental loading of roots in tension or in compression. Our model shows that it is important to consider the forces held by roots in a way that is entirely different than done thus far. Our work





quantifies the contribution of roots in tension and compression which now finally permits to analyze more realistically the role of root reinforcement during the triggering of shallow landslides.

# 1 Introduction

Shallow landslides are hillslope processes that play a key role in shaping landscapes in forested catchments (Istanbulluoglu and Bras, 2005; Sidle and Ochiai, 2006). In some regions, shallow landslides are the dominant regulating mechanisms by which soil is delivered from the hillslope to steep channels or fluvial systems (Jakob et al., 2005). The magnitude and intensity of these phenomena also has important societal impacts, both in the long (landscape evolution and soil resource availability (Istanbulluoglu and Bras, 2005; Montgomery, 2007)) and short term (risks due to landslides, debris flows and sediment transport, water quality, soil productivity (Wehrli et al., 2007; Hamilton, 2008)).

On long time scales, shallow landslides are important geomorphic processes shaping landscapes of both vegetated and non-vegetated basins. For vegetated basins, the spatio-temporal distribution of root reinforcement has a major impact on the dynamic of sediment transport at the catchment scale (Sidle and Ochiai, 2006) and on the availability of productive soil, a key resource for human needs. At the hillslope scale, the presence of vegetation generally increases soil thickness, lowering the frequency of landsliding events but increasing their magnitudes (Amundson et al., 2015). At the catchment scale, vegetation causes slopes to steepen and sediment mobilization is then often dominated by deep landslides driven by fluvial incision (Larsen and Montgomery, 2012). The influence of shallow landslides on shaping the landscape on long time scales is, in part, masked by continuously-changing factors influenced by human activities, climate change, and other disturbances such as storms, fires, etc. Under these constant disturbances soils never reach an equilibrium state that would otherwise require between 10 and 1000 years. Nevertheless, the presence of soils on steep slopes are a necessary condition for preserving important functions of mountain environments, such as water supply, nutrient production, biodiversity, landscape aesthetics, and cultural heritage.

While soil as a resource is gaining increasing attention in the context of global sustainable development (Nature Editorial, 2015), risks related to shallow landslides and to processes linked to them (debris flows, bedload transport, large wood transport during floods) as well as the availability of quality water are issues that impact human societies in the short term (Miura et al., 2015) particularly in mountainous regions. Water quality is linked to shallow landslides because sediments mobilized by landslides are transported as suspended sediments in streams.

While sustainable resource management in forestry and in agriculture targets to keep the frequency of shallow landslide events to pseudo-equilibrium conditions at the catchment scale and to reduce the overall erosion rate (Li et al., 2016), disturbances such as those due to human activities may lead to a rapid and dramatic increase of shallow landslide frequency and magnitude. For instance, deforestation and intensive agriculture may lead to an increase of the overall erosion rate by one order of magnitude. Marden (2012) reports that in the 17 km$^2$ catchment of Waipaoa (New Zealand), erosion rate increased from 2.7 to 15 Mt year$^{-1}$ after deforestation and conversion of slopes to pasture land. In this new environment, shallow landslides contribute ~60% of the sediment yield of the Waipaoa river during floods and 10 to 20% of total erosion. Similar conditions occurred in the European Alps until the first half of the 20th century that led to a considerable increase in erosion rates (Mar-




iotta, 2004). Meusburger and Alewell (2008) report that, in a catchment in the central Alps, the increase of landslide area by 92% within 45 years was likely due to dynamic factors like climate and land-use changes and had a decisive influence on landslide patterns observed today.

Risks due to shallow landslides are associated with different types of phenomenon ranging from hillslope debris flows (example of process causing a direct risk to infrastructures and individuals) to various channel processes such as large sediment transport during floods, wood debris transport, channelized debris flows, etc. (examples of processes causing an indirect risk to infrastructures and individuals). It is estimated that landslides triggered by heavy rainfall cause damages upwards of several millions each year and more than 600 fatalities per year (Sidle and Ochiai, 2006).

Next to the constellation of factors well-known to influence the triggering of shallow landslides, vegetation has been recognized to play an important role (Sidle and Ochiai, 2006; Schwarz et al., 2010c; McGuire et al., 2016) and its function is considered an important component of ecosystem services provided in mountain regions. The importance of the effects of vegetation is, in some cases, recognized at a political level. For instance, the global forest area managed for protection of soil and water is 25% of all global forested areas (Miura et al., 2015). In Switzerland, protection forests occupy more than 50% of all forested areas (Wehrli et al., 2007). Moreover, bio-engineering measures are often considered an important part of integrated risk management and disaster risk reduction strategies. The management of such protection forests and bio-engineering measures needs quantitative tools to optimize the effectiveness of such important ecosystem services for society. The formulation of such tools needs to be based on quantitative methods applicable to a large range of situations. Moreover, these methods need to consider different time and spatial scales at which vegetation influences processes. To put the motivation for the present work in the appropriate context, we shortly summarize the effects of vegetation on long and short term processes.

In the long term, the presence of vegetation

- increases soil production rates through mechanical and chemical processes (Wilkinson et al., 2005; Phillips et al., 2008) (100–1000 years);

- increases soil residence time on hillslopes due to root reinforcement and protects against runoff erosion (Istanbulluoglu and Bras, 2005) (10–100 years). Note that in the case of natural or human driven disturbances, the response of the system (i.e. root decay) is of the order of a few years (Vergani et al., 2016);

- enhances soil diffusion rates on hillslopes due to tree wind-throw (Pawlik, 2013; Roering et al., 2010), root mounds (Hoffman and Anderson, 2014), and biological activity (Gabet and Mudd, 2010) (100–1000 years).

In the short term, vegetation mainly influences root reinforcement and regulates water fluxes. At the hillslope scale, the hydrological effects of vegetation are assumed to play a small role on slope stability compared to the contribution of root reinforcement (Sidle and Bogaard, 2016; Sidle and Ziegler, 2017). At the catchment scale, however, the regulation of water fluxes may have important implications for the stability of those slopes that drain large areas, particularly for short and intense rainfall events.

Root are considered the hidden half of plants due to the difficulties in characterizing and quantifying their distribution and mechanical properties. In slope stability, the process of root reinforcement remains hidden because direct observations have




not yet been made on steep hillslopes. Field and laboratory experiments (e.g. Zhou et al., 1998; Ekanayake and Phillips, 1999; Roering et al., 2003; Docker and Hubble, 2008) generally explore only a small part of the complex root reinforcement mechanisms.

## 1.1 Mechanisms of root reinforcement

Many studies have highlighted the importance of roots and their mechanical properties for the stabilization of hillslopes (e.g. Schwarz et al., 2015), but usually, only basal root reinforcement is considered. In a wider analysis of how roots reinforce soil, three different mechanisms of root reinforcement must be recognized:

- Basal root reinforcement acts on the basal shear surface of the landslide. This is the most efficient mechanism, if present. In many cases, however, this mechanism is absent because the position of the failure surface is deeper than the rooting
zone.

- Lateral root reinforcement acts on lateral surfaces of the landslide. The magnitude of the contribution of this mechanism depends on the type of deformation of the landslide mass. If the landslide behaves as a rigid mass, lateral reinforcement may act almost simultaneously along all the edges of the sliding mass (in tension, shear, and compression). In cases where there is differential deformation of the soil mass, this leads to the progressive activation of lateral reinforcement,
first in tension at the top of the landslide, and then in compression on the toe at the end of the triggering . The magnitude of lateral root reinforcement depends on the spatial distribution of the root network.

- Roots stiffen the soil mass. The presence of roots in the soil increases the macroscopic stiffness of the rooted soil mass leading to a larger redistribution of forces at the scale of the hillslope through small deformations. This mechanism increases the effects of the previous two (basal and lateral root reinforcement).

On top of these considerations on root reinforcement mechanisms acting on a single landslide, it is important to highlight that those mechanisms assume different meaning when considering the more global context of landslide processes at the catchment scale. Specifically, the effects of root reinforcement on landslide processes are considered limited by:

- The magnitude of root reinforcement (a function of forest structure and tree species composition). Root reinforcement needs to reach values of the order of a few hPa in order to be significant.

- The heterogeneity of root distribution (tree species, topography, local soil condition, etc.). Root reinforcement must be active in specific places and at specific times to have any effect on slope stability: mean values of apparent cohesion across the entire hillslope are not representative and not sufficient for considering the specifics of actual root reinforcement effects.

- The depth of the landslide shear surface (effects of basal root reinforcement). The deeper the shear surface is, the less
important the effect of basal root reinforcement is.




- The length and volume of the landslide (lateral root reinforcement and buttressing/arching mechanisms and stiffening effects). The larger the landslide is, the lower are the effects of lateral root reinforcement.

In order to characterize the efficacy of roots for slope stabilization, a spatio-temporal quantification of root reinforcement is needed.

## 1.2 Models of root reinforcement

Methods for the quantification of different types of root reinforcement mechanisms went through a succession of models in the last few decades, starting with the assumption of the simultaneous breakage of all roots (Wu et al., 1979; Waldron and Dakessian, 1981) to the application of fiber bundle models that consider the progressive failures of roots (Pollen and Simon, 2005; Schwarz et al., 2010a; Cohen et al., 2011). Fiber bundle models may be differentiated on the basis of the type of loading, whether it is by stress (Pollen and Simon, 2005) which does not allow for the calculation of displacement, or by strain (Schwarz et al., 2013; Cohen et al., 2011), which does. We enumerate below some aspects of root reinforcement models important for slope stability.

- Breakage versus slip out. Field observations show that in tree-root bundles, the dominating failure mechanism of roots is by breakage (Schwarz et al., 2012a). Slippage is limited to small roots that usually contribute only a small fraction of the total root reinforcement. For this reason, numerical models usually assume that all roots fail by breaking (Schwarz et al., 2013; Cohen et al., 2011).

- The contribution of root reinforcement must be differentiated between different types of stress conditions: tension, compression, and shearing. While most of the literature has focused on the shear behavior of rooted soils (e.g. Docker and Hubble, 2008), some works have investigated the contribution of root reinforcement under tension (Zhou et al., 1998; Schwarz et al., 2010a, 2011; Cohen et al., 2011) and compression (Schwarz et al., 2015). In general the contribution of maximum root reinforcement under tension and shearing is of the same order of magnitude, whereas under compression the contribution of roots is about one order of magnitude smaller. However, roots contribute significantly to increase the stiffness of soil under compression. This may play an important role in the re-distribution of forces during the triggering of a shallow landslide (Schwarz et al., 2015).

- The mechanical interactions of neighboring roots in a bundle is usually neglected. Giadrossich et al. (2013) showed with laboratory experiments that the failure mechanisms of single roots is influenced by neighboring roots only at high root density that are usually reached only near tree stems (0–0.5 m).

- The mechanical and geometrical variability of roots was recently considered using survival functions (Schwarz et al., 2013) that represent the complexity of several factors contributing to the variable stress-strain behavior of roots. Specifically, these factors are root tortuosity (Schwarz et al., 2010a), root-soil mechanical interactions (Schwarz et al., 2011), and position of root breakage along the root. Pulled roots break at different distances from the point of force application because of branching, root geometry, changes in root diameter due to soil properties, presence of stones, etc.



– The spatial and temporal heterogeneity of root reinforcement is related to several factors such as topography, soil water content, soil disturbances, resistance and resilience of forest cover to disturbances, animal browsing, etc. (Schwarz et al., 2012a; Vergani et al., 2016)

### 1.3 Implementation of root reinforcement in slope stability models

In view of the importance of root reinforcement and of shallow landslides to landscape evolution and to human societies, mechanistic models that fundamentally understand the processes linked to the triggering of shallow landslide and the influence of root reinforcement on it are needed. In the large majority of cases, slope stability models add apparent cohesion to the soil to simulate root reinforcement (e.g. Milledge et al., 2014; Bellugi et al., 2015; Hwang et al., 2015). Few models includes the effects of root distribution heterogeneity (Stokes et al., 2014), and none consider the stress-strain behavior of root reinforcement

and the strength of roots in compression. Recent field observations show that shallow landslide triggering mechanisms are characterized by differential deformation that indicates localized loading of soils in tension, compression, and shear (Schwarz et al., 2012a). These observations contradict common assumptions used in models until now. However, the direct coupling of these different root reinforcement mechanisms, and their mobilization during the triggering of shallow landslides, has not yet been made.

Here we present a new model for shallow slope stability calculations that specifically considers these important effects. To fully understand the mechanisms of shallow landslide triggering, a complete re-evaluation of the traditional apparent cohesion approach is required. To do so, it is important to consider the forces held by roots in a way that is entirely different than done thus far. Moreover, measurements and models indicate that the assumptions of constant elasticity and homogeneous root properties, as applied in typical finite element geotechnical model, cannot reproduce the mechanisms leading to the triggering

of forested slope failures.

The SOSlope (for Self-Organized Slope) model presented here, fills this gap by developing a mechanistic model for predicting shallow landslide sizes across landscapes, considering the effects of root reinforcement in a detailed quantitative manner (spatio-temporal heterogeneity of root reinforcement). The SOSlope model allows to explore the activation of root reinforcement during the triggering process and helps to shed light on the contribution of roots to the slope stability. The SOSlope model

is used in this work to test the following main hypotheses:

– Both tension and compression forces are efficient in stabilizing slopes and have higher effectiveness when combined.

– Weak zones in the root network (Schwarz et al., 2010b, 2012a) determine the effectiveness of root reinforcement at the slope scale if no basal reinforcement is present.

– Coarse roots dominate reinforcement and its efficacy, when present.



## 2 The SOSlope model

### 2.1 General framework

SOSlope is a hydro-mechanical model of slope stability that computes the factor of safety on a hillslope discretized into a two-dimensional array of blocks connected by bonds. Bonds between adjacent blocks represent mechanical forces acting across the

blocks due to roots and soil (Cohen et al., 2009). These forces can either be tensile or compressive depending on the relative displacements of the blocks. A digital elevation model (DEM) is used to divide the hillslope into squares in plan view where the centers of the squares are points of the DEM (Fig. 1). Three dimensional blocks are created by extruding the squares to the bottom of the soil layer along the vertical. The center of mass of a block is connected to the four lateral blocks by four force bonds (Fig. 1). Initially, bond forces between blocks are set to zero. Rainfall onto the slope will increase the mass and

decrease the soil shear strength of the blocks. At each time step, the factor of safety is calculated for each block using a force balance (resistive force over active force, see equations below). If the factor of safety of one or more blocks is less than one, those blocks are moved in the direction of the local active force (defined below) by a predefined amount (usually 0.1 mm) and the factor of safety is recalculated for all blocks. Because of the relative motion between blocks that have moved and blocks that remain stationary, mechanical bond forces between blocks are no longer zero and the force balance changes. This relative

motion triggers instantaneous force redistributions across the entire hillslope similar to a self-organized critical (SOC) system of which the spring-block model (Bak et al., 1988; Hergarten and Neugebauer, 1998; Cohen et al., 2009) is a subset. Looping over blocks and moving those that are unstable is repeated until all blocks are either stable (factor of safety greater or equal to 1) and the system reaches a new equilibrium, or some blocks have failed (their displacements are greater than some set value, usually a few meters), triggering a landslide.

### 2.2 Factor of safety

The factor of safety for each block is calculated as the ratio of resistive to active forces. Resistive forces include the soil basal shear strength and the strength of roots that cross the basal slip surface, assumed to be located at the bottom of the soil layer. The active forces include the gravitational driving force due to the soil mass and the push or pull forces between blocks that include the effects of soil and root tension and compression. These later forces are the bond forces between the blocks described

above. Including all these forces in a force balance yields the factor of safety

$$\text{FS} = \frac{F_s + F_r}{\left\| \boldsymbol{F}_d + \sum_{j=1}^{4} \boldsymbol{F}_j \right\|}, \tag{1}$$

where $F_s$ is the soil basal resistive force that includes soil cohesion and friction, $F_r$ is the basal root resistance, $\boldsymbol{F}_d$ is the driving force vector due to gravity, and $\boldsymbol{F}_j, j = 1, \cdots, 4$, are the 4 bond vector forces that quantify soil and root tension or compression between the block and its four neighbors. The vertical bars in the denominator denote the norm of a vector. This factor of safety

is calculated for each block but an index for the block number is not included so as not to clutter the equations.





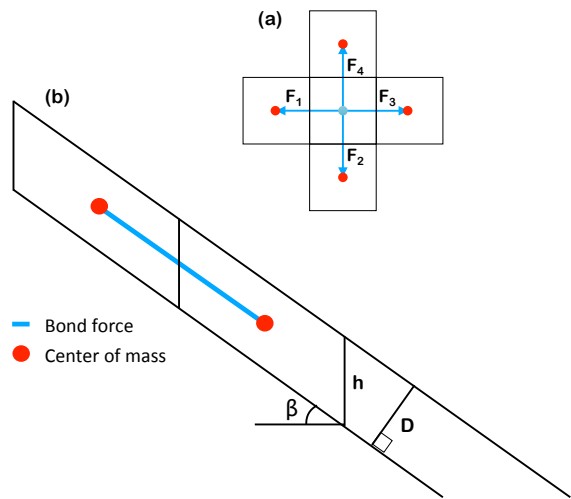

**Figure 1.** (a) Plan view of discretized cell with its four neighbors showing bond forces. (b) Vertical section across neighboring cells showing the center of mass of cells and the location of the connecting bond. $\beta$ is the surface slope and $h$ and $D$ are the thicknesses of soil down to the basal surface, measured vertically and perpendicular to the surface, respesctively.

Soil basal resistance is

$$F_s = A\tau_b, \tag{2}$$

where $A$ is the surface area of the block along the failure surface and $\tau_b$ is the basal shear stress (described below). In the present model, we set $F_r = 0$, focusing on lateral root reinforcement. This is justified in many cases where the depth of the slip surface is 1 meter or greater and very few roots are present (e.g. Bischetti et al., 2005; Tron et al., 2014). Basal root reinforcement can easily be added using a formulation similar to lateral root reinforcement (discussed below) with values of root reinforcement a function of the shear displacement and the density of roots crossing the slip surface.

The driving force is

$$\boldsymbol{F}_d = \gamma D A \hat{\boldsymbol{t}}, \tag{3}$$

where $\gamma$ is the specific weight of the wet soil, $D$ is the depth to the shearing surface, perpendicular to slope, and $\hat{\boldsymbol{t}}$ is the unit tangent to the slope in the direction of the maximum slope. The specific weight of the wet soil is calculated based on water content and solid fraction, i.e.,

$$\gamma = (\rho_s \phi^s + \rho_w \theta)g, \tag{4}$$




where $\rho_s$ and $\rho_w$ are the solid (grain) and water densities, respectively, $\phi_s$ is the solid volumetric fraction, $\theta$ the volumetric water content, and $g$ is gravity.

Bond forces are given by

$$\boldsymbol{F}_j = \left(F_j^{\text{soil}} + F_j^{\text{roots}}\right) \hat{\boldsymbol{b}}_j, \quad j = 1, \cdots, 4, \tag{5}$$

where $F_j^{\text{soil}}$ and $F_j^{\text{roots}}$ are the soil and root components of the four bond forces, respectively, and $\hat{\boldsymbol{b}}_j$ are unit vectors along the bond axes pointing outward of the block. These quantities are detailed below.

## 2.3   Bond forces due to roots

The force in bond $j$ between a block and its neighbor due to roots ($F_j^{\text{root}}$) depends on four factors: the root density and the root-diameter distribution at the bond center, the strength of roots which depends on root diameter, and the change in length

(elongation) of the bond with respect to its initial length. This force is computed using the Root Bundle Model (RBM) of Schwarz et al. (2013) with Weibull statistics, called RBMw. For the sake of completeness, the full details of the model are given below.

### 2.3.1   Root density and root-diameter distribution

Roots are binned according to their diameters in 1-mm size bins from 0.5 mm to an upper limit given by data. A bin is usually

referred to as a root-diameter class, with $\phi_i$ denoting the mean root diameter of class $i$, $i = 1, \cdots, i_{\text{max}}$. At each of the four faces of a block, the total number of roots for each root-diameter class $i$ that crosses a face $j$ is the sum of the number of roots for that root-diameter class from each surrounding tree in the stand. Summing roots from each tree implies no competition for resources. Following the empirical model of Schwarz et al. (2010a) in its version described by Giadrossich et al. (2016), the number of roots depends on the distance of the face center to the tree trunks, the tree trunks diameters, and the tree specie.

For simplicity all trees in the stand are assumed to belong to the same specie. The model assumes a linear allometric relation between trunk size and root density, a power-law decay of root density with distance from the tree trunk, and a logarithmic decrease of root density with root-diameter size. The number of roots of class diameter $\phi_i$ crossing face $j$ is

$$N_{\phi_i}^j = A_j \sum_{k=1}^{T} \rho_k^j \left[1 - \frac{\ln\left(1 + \min(\phi_i, \phi_k^{\text{max}})/\phi_o\right)}{\ln\left(1 + \phi_k^{\text{max}}/\phi_o\right)}\right] \left(\frac{\phi_i}{\phi_o}\right)^\gamma, \tag{6}$$

where $A_j$ is the surface area of face $j$, $T$ is the number of trees in the stand (more specifically the number of trees whose roots

reach face $j$ of the cell), and $\rho_k^j$ is the density of fine roots of tree $k$ for face $j$. This later quantity is given by

$$\rho_k^j = \frac{N_k}{d_k^{\text{max}} 2\pi d_k^j} \left[\frac{\max(0, d_k^{\text{max}} - d_k^j)}{d_k^{\text{max}}}\right], \tag{7}$$

where $N_k$, the total number of fine roots of tree $k$, is

$$N_k = \mu\pi \left(\frac{\phi_k}{2}\right)^2, \tag{8}$$



$d_k^{\mathrm{max}}$, the maximum rooting distance for tree $k$, is

$$d_k^{\mathrm{max}} = \psi\,\phi_k, \tag{9}$$

and $\phi_k^{\mathrm{max}}$, the maximum root diameter class of tree $k$, is

$$\phi_k^{\mathrm{max}} = \max\left(0, \frac{d_k^{\mathrm{max}} - d_k^j}{\eta}\right). \tag{10}$$

In these equations, $\phi_o = 1\ \mathrm{mm}$ is the size of the smallest root diameter class, $d_k^j$ is the distance between face $j$ and tree $k$, and $\phi_k$ is the tree diameter (usually diameter at breast height or simply DBH). This model contains four fitting parameters ($\mu$, $\eta$, $\psi$, and $\gamma$) that must be determined from data (Giadrossich et al., 2016; Schwarz et al., 2016).

### 2.3.2 Root mechanical forces

Roots are assumed elastic in both tension (Schwarz et al., 2013) and compression (Schwarz et al., 2015). The linear elastic force in a root is expressed using a spring constant (i.e., Hooke's law) that depends on the root diameter class. For a root in diameter class $i$ on bond $j$, that elastic force is

$$F_{i,j}^{\mathrm{E}}(\phi_i, x_j) = k_i^{\mathrm{E}}\,x_j, \tag{11}$$

where the superscript E indicates either tension (E = T) or compression (E = C) and $x_j$ is the elongation of the bond from its initial length (positive for tension, negative for compression). Based on data (e.g. Schwarz et al., 2013, 2015) we assume the spring constant depends linearly on root diameter, i.e.,

$$k_i^{\mathrm{E}} = k_0^{\mathrm{E}} + k_1^{\mathrm{E}}\,\phi_i, \tag{12}$$

with $k_0^{\mathrm{E}}$ and $k_1^{\mathrm{E}}$ constants to be determined from data. Other formulations based on a power-law relation can also be used Giadrossich et al. (2016).

The variability of root bio-mechanical properties (e.g. maximum tensile or compressive strength, elastic moduli in tension or compression) due to the presence of biological or geometrical weak spots is handled probabilistically. The probability of failure of a root in tension (or in compression) is captured by multiplying the elastic force by a Weibull survival function ($S$) that depends on a dimensionless bond elongation. Then, the total root-bond force is obtained by summing over all roots of each diameter class, i.e.,

$$F_j^{\mathrm{roots}}(x_j) = \sum_{i=1}^{i_{\mathrm{max}}} N_{\phi_i}^j\,F_{i,j}^{\mathrm{E}}(\phi_i, x_j)\,S_{i,j}^{\mathrm{E}}(\xi_{i,j}), \tag{13}$$

where $N_{\phi_i}^j$ is given by Eq. 6, $F_{i,j}^{\mathrm{E}}$ by Eq. 11,

$$S_{i,j}^{\mathrm{E}}(\xi_{i,j}) = \exp\left[-\left(\frac{\xi_{i,j}}{\lambda^{\mathrm{E}}}\right)^{\omega^{\mathrm{E}}}\right], \tag{14}$$





and

$$\xi_{i,j} = \frac{k_i^{\mathrm{E}} x_j}{F_{i,\max}^{\mathrm{E}}}, \tag{15}$$

where $\lambda^{\mathrm{E}}$ and $\omega^{\mathrm{E}}$ (E = T or C) are two scale and two shape parameters to be determined from field or laboratory experiments (see Schwarz et al., 2013, 2015, for details). $F_{i,\max}^{\mathrm{E}}$ is the maximum force held in a root at breakage (in tension) or at the critical

buckling condition (in compression, see Schwarz et al., 2015) for a root of diameter $\phi_i$ and is given by the commonly used power-law equation

$$F_{i,\max}^{\mathrm{E}} = F_o^{\mathrm{E}} \left( \frac{\phi_i}{\phi_o} \right)^{\alpha^{\mathrm{E}}}, \tag{16}$$

with $\alpha^{\mathrm{E}}$ the power-law exponent and $F_o^{\mathrm{E}}$ a pre-exponential factor for tension or compression (E = T or C). The scaling of the displacement with the maximum strength of a root eliminates the effect of root diameter on maximum displacement. Similarly,

the parameter $\lambda^{\mathrm{E}}$ scales the root strength variability to the root diameter. Equation 13 has a maximum ($F_{j,\max}^{\mathrm{roots}}$) called the maximum root reinforcement and occurs at a bond elongation $x_{j,\max}$.

## 2.4 Bond forces due to soil

The soil bond force ($F_j^{\mathrm{soil}}$, Eq 5) depends on whether the soil is in tension or in compression. For tension, we assume that resistance scales with soil apparent cohesion (including the effects of suction stress for unsaturated soils) as a function of

displacement using a logarithmic function (Win, 2006)

$$F_j^{\mathrm{soil, \, T}} = \begin{cases} c_a \, W \, D \left( 1 - \frac{\log(1+\varepsilon_j L_j)}{\log(1+\varepsilon_{\max}^{\mathrm{T}} L_j)} \right), & \varepsilon_j < \varepsilon_{\max}^{\mathrm{T}}, \\ 0, & \varepsilon_j \geq \varepsilon_{\max}^{\mathrm{T}}, \end{cases} \tag{17}$$

where $c_a$ is the apparent cohesion, $\varepsilon_{\max}^{\mathrm{T}}$ is a strain threshold above which soil loses any tensional resistance, and $L_j$ is the length of bond $j$. In compression, following the work of Schwarz et al. (2015) we assume that the soil compressional resistance is mobilized across the shear plane that forms during the failure of a downslope wedge, similar to the earth pressure force in

the geotechnical engineering literature that develops during passive state when a retaining wall moves downslope toward the adjacent backfill (e.g. Milledge et al., 2014). According to Schwarz et al. (2015), the mobilized force on the downslope wedge scales with the maximum passive earth pressure force $F_p$ and with the displacement, i.e.,

$$F_j^{\mathrm{soil, \, C}}(x_j) = -F_p \, W \, P_{w_1}(x_j) \, S_{w_2}(x_j) \tag{18}$$

where

$$F_p = K_{p\gamma} \rho g \frac{D^2}{2} + K_{pc} c' D, \tag{19}$$

and $K_{p\gamma}$ and $K_{pc}$ are the passive earth pressure coefficients due to soil weight and to cohesion, respectively, obtained from a fitting of equations given in Soubra and Macuh (2002), $c'$ is effective soil cohesion, and $P_{w_1}$ and $S_{w_2}$ are the Weibull cumulative





density and the Weibull survival functions, respectively, given by

$$P_{w_1}(x_j) = 1 - \exp\left[-\left(\frac{x_j}{\mu_1}\right)^{\kappa_1}\right], \tag{20}$$

and

$$S_{w_2}(x_j) = \exp\left[-\left(\frac{x_j}{\mu_2}\right)^{\kappa_2}\right], \tag{21}$$

with $\mu_1$, $\kappa_1$, $\mu_2$, and $\kappa_2$ four parameters determined from compression experiments. The first Weibull function, $P_{w_1}$, serves
to scale the maximum passive earth pressure force with displacement during initial block motion while the second one, $S_{w_2}$,
reduces that same force as the wedge is overridden by the block and the failure surface area of the slip plane decreases (see
Schwarz et al., 2015, for details). We neglect the active earth pressure force on upstream faces of cells because the magnitude
of the active force is small in comparison to other forces.

## 2.5   Hydrological triggering

Rainfall-triggered shallow landslides can fail under saturated conditions during increases of pore-water pressure and/or loss of
suction under unsaturated conditions (Lu and Godt, 2013). Our objective here is not to reproduce the detailed physical mecha-
nisms by which changes in subsurface hydrology trigger a landslide but to develop a simple empirical model that realistically
mimics observed changes in pore-water pressure and water content during rainfall infiltration. Although diverse hydrologic
triggers have been observed and described (e.g. Reid et al., 1997; Iverson, 2000), here we use, as a representative example
for the hydrological conditions triggering a shallow landslide in our model, pore-pressure measurements during the artificial
triggering of the Ruedlingen shallow landslide experiment in Switzerland (Askarinejad et al., 2012; Lehmann et al., 2013).
Data from Lehmann et al. (2013) indicate that high pore-water pressures were attained relatively quickly and remained steady
across the slope long before failure occurred, and that the decrease in the standard deviation of the water saturation prior to
failure indicated an increase in the connectivity of water saturated regions that reduced soil shear strength across the full length
of the slip surface leading to failure. Other data in different localities (e.g. Matsushi et al., 2006; Bordoni et al., 2015) have also
shown high, steady pore-water pressure prior to failure. Because our model focuses on the effects of roots and soil strength on
slope stability rather than on the details of hydrologic triggering, we choose a simplified, empirical, dual-porosity model for our
slope hydrology. Our objective is only to reproduce reasonable pore-water pressure distribution and water content evolution in
both the matrix and the preferential flow domains, but not to model the physics of evolving subsurface hydrology. The model
embodies the rapid increase in positive pore-pressure in a preferential flow domain (representing macropores) and the slow
decrease of suction in the soil matrix caused by slow water transfer from the macropores to the matrix. This decrease of suction
is the equivalent of the increasing connections of water saturated regions represented by the decrease in the standard deviation
of water saturation observed by Lehmann et al. (2013) that eventually caused slope failure in the Ruedlingen experiment.
We assume that water flow in soils during a rainfall event is a combination of slow matrix flow (also called immobile water
with capillary number lower then 1) and fast preferential flow (mobile water, capillary number higher than 1) (Sidle and Ochiai,
2006; Beven and Germann, 2013). While slow matrix flow influences the change in suction stress, the fast preferential flow





directly influences pore-water pressure in the macropores. Our formulation of this concept is empirical and is a simplification of the more common dual-porosity models that employ two flow equations (e.g. Richards' equation) that exchange moisture between the two domains, and mixture equations for water content, hydraulic conductivity, rainfall partitioning based on the volumetric ratio of the fast and slow flow domains (e.g. Gerke and van Genuchten, 1993; Shao et al., 2015). In accord with

continuum mixture theory for effective stress (e.g. Borja and Koliji, 2009), we write the mean pore-water pressure of the soil (matrix + macropores), $\bar{p}$, as

$$\bar{p} = \psi^1 \bar{p}_1 + \psi^2 \bar{p}_2 \tag{22}$$

where $\psi^1$ and $\psi^2$ are the pore fractions along the potential failure surface of the landslide of the matrix and the macropores, respectively (volume of pore in matrix or macropores over total pore volume, with indices 1 for matrix and 2 for macropores)

with $\psi^1 + \psi^2 = 1$, and where $\bar{p}_i, i = 1, 2$, are the matrix and macropores intrinsic mean pore pressures. Pore fractions $\psi^1$ and $\psi^2$s are related to the partial porosities of the matrix and the macropores, $\phi^1$ and $\phi^2$, respectively, by

$$\psi^i = \frac{\phi^i}{n}, \quad i = 1, 2, \tag{23}$$

where $\phi^1 + \phi^2 = n$, $n$ the total porosity of the soil. The solid volume fraction of the matrix (macropores have only pore space) is $\phi^s = 1 - \phi^1 - \phi^2 = 1 - n$. The superscripts and subscripts in these equations and in equations below refer to partial and

intrinsic quantities, respectively. Partial and intrinsic water content of the matrix and macropores are related as follows:

$$\theta^1 = (\phi^s + \phi^1)\theta_1, \tag{24}$$

$$\theta^2 = \phi^2 \theta_2, \tag{25}$$

where $\theta^1$ and $\theta^2$ are the partial water contents of phase 1 and 2 (volumetric water content of phase 1 or 2 over total soil

volume) and $\theta_1$ and $\theta_2$ are the intrinsic water contents of each phases (volumetric water content of phase $i$ over volume of phase $i$, $i = 1, 2$). At saturation $\theta^2 = \phi^2$ since the macropore phase contains only void space and thus $\theta_2 = 1$. The total water content of the soil is

$$\theta = \theta^1 + \theta^2, \tag{26}$$

and is used in Eq. 4 to compute the soil specific weight. Equations similar to Eq. 26 can be written for saturated and residual

water contents.

We assume that the time evolution of the intrinsic pore-water pressure in the macropores, $\bar{p}_2$, and of the partial water content in both the macropore ($\theta^2$) and the matrix phases ($\theta^1$) can be modeled using cumulative distribution functions. For the macropore phase, we write

$$\bar{p}_2(t) = p_{\max} F(t^\star, \mu_p, \sigma_p), \tag{27}$$





and

$$\theta^2(t) = \phi^2 \left[ \theta_2^r + (1 - \theta_2^r) F(t^\star, \mu_p, \sigma_p) \right], \tag{28}$$

where $p_{\max}$ is a constant here but ultimately depends on rainfall infiltration rate and upstream contributing area (e.g. Montgomery and Dietrich, 1994), $t^\star$ is a dimensionless time, $F$ is the normal cumulative distribution function with mean $\mu_p$ and standard deviation $\sigma_p$, and $\theta_2^r$ is the intrinsic residual water content for the macropores (we have used the fact that the intrinsic saturated water content $\theta_2^s = 1$ since macropores have no solid fraction). For the water content in the matrix we assume that

$$\theta^1(t) = (\theta_o - \phi^2) + (\theta_s - \theta_o) F_{\text{fold}}(t^\star, \mu_\theta, \sigma_\theta), \tag{29}$$

where $\theta_o$ and $\theta_s$ are the soil initial and saturated water contents, respectively, and $F_{\text{fold}}$ is the folded normal cumulative distribution with mean $\mu_\theta$ and standard deviation $\sigma_\theta$. The pore-water pressure in the matrix is given by (Borja and Koliji, 2009)

$$\bar{p}^1(t) = S_e^1 p_1, \tag{30}$$

where $p_1$ is the intrinsic pore-water pressure in the matrix and $S_e$ is the equivalent degree of saturation (also called effective saturation) in the matrix. Following Lu et al. (2010), we have used the equivalent degree of saturation ($S_e$) in Eq. 33 instead of the more commonly used degree of saturation. Under unsaturated condition, $\bar{p}^1$ is a matrix suction stress (Lu et al., 2010). The equivalent degree of saturation in the matrix is defined as

$$S_e^1 = \frac{\theta_1 - \theta_1^r}{\theta_1^s - \theta_1^r}, \tag{31}$$

where $\theta_1^r$ and $\theta_1^s$ are the intrinsic residual and saturated water content of the matrix phase with $\theta_1^s = \phi^1$. Using Eqs. 24 and 25, and equations for the residual and saturated water content equivalent in for to Eq. 26, Eq. 31 can be rewritten as

$$S_e^1 = \frac{\theta^1 - \theta_r}{\theta_s - \phi^2 - \theta_r}, \tag{32}$$

where $\theta^1$ is given by Eq. 29. Using van Genuchten formulation (Van Genuchten, 1980), we can write the suction stress as (Lu et al., 2010)

$$\bar{p}^1(t) = -\frac{S_e^1}{\alpha_{vg}} \left( \left( S_e^1 \right)^{\frac{n_{vg}}{1 - n_{vg}}} - 1 \right)^{\frac{1}{n_{vg}}}, \tag{33}$$

where and $\alpha_{vg}$ and $n_{vg}$ are the soil parameters.

Pore-water pressure in the macropores (Eq. 27), matrix water content (Eq. 29), matrix suction (Eq. 33), and mean pore-water pressure (Eq. 22) are computed at each block of the domain at each time step. The dimensionless time $t^\star$ in these equations is time scaled with the characteristic time for reaching steady state ($t^\star = t/t_{ss}$). Figure 2 illustrate the model behavior for parameters shown in Table 1. The standard deviations are chosen so that macropore water pressure reaches its maximum before matrix water content, to mimic, but not reproduce, the behavior observed by Lehmann et al. (2013).





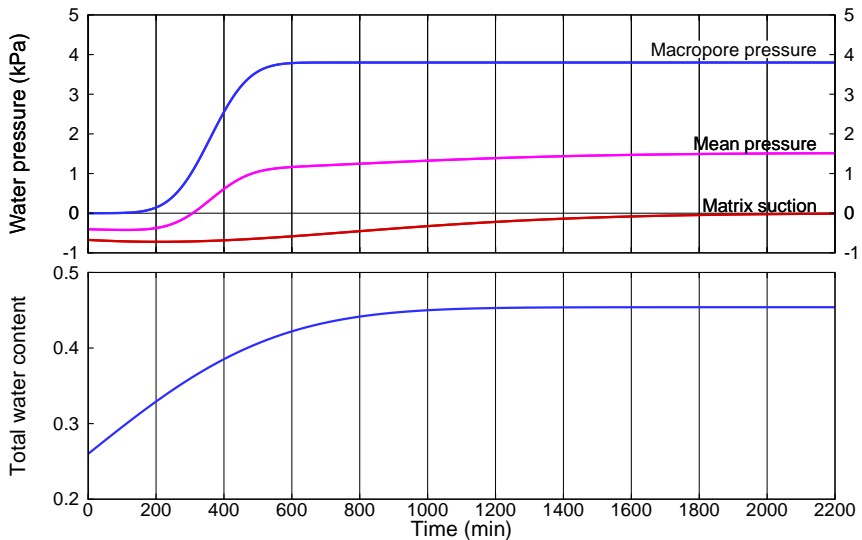

**Figure 2.** Time evolution of pore-water pressures and water content for the dual-porosity model.

**Table 1.** Hydrological parameters used in all simulations.

| Variable | Value |
| --- | --- |
| $t_{ss}$ | 720 min |
| $\mu_p$ | 0.5 |
| $\sigma_p$ | 0.125 |
| $\mu_\theta$ | 0.0 |
| $\sigma_\theta$ | 0.6 |

## 2.6 Basal shear stress

Basal shear resistance along the slip surface is calculated using the Mohr-Coulomb failure criterion including contributions from both the suction stress and the pore-water pressure using the mean pore-water pressure $\bar{p}$ of Eq. 22, i.e.,

$$\tau_b = c' + [\sigma_n - \bar{p}] \tan\phi, \tag{34}$$

5  where $\sigma_n$ is the normal stress and $\phi$ is the soil friction angle.





## 3 Data

### 3.1 Soil

Mechanical soil parameters from Schwarz et al. (2013, 2015) and other parameters used in simulations are listed in Table 2. Figure 3 shows the soil strength in tension and compression (positive and negative values of displacement, respectively) for different soil thicknesses.

**Table 2.** Soil parameters used in all simulations.

| Variable | Value | Units |
|---|---|---|
| $\rho_s$ | 1700 | kg m$^{-3}$ |
| $\rho_w$ | 1000 | kg m$^{-3}$ |
| $c'$ | 500 | Pa |
| $\phi$ | 31 | degrees |
| $D$ | 1 | m |
| $\varepsilon_{\max}^{\mathrm{T}}$ | 0.003 | |
| $\mu_1$ | 0.58 | m |
| $\kappa_1$ | 0.07 | |
| $\mu_2$ | 2.00 | m |
| $\kappa_2$ | 0.25 | |
| $\theta_s = n$ | 0.46 | |
| $\theta_r$ | 0.082 | |
| $\theta_o$ | 0.26 | |
| $\psi^1$ | 0.4 | |
| $\psi^2$ | 0.6 | |
| $p_{\max}$ | 3800 | Pa |
| $n_{vg}$ | 3.3 | |
| $\alpha_{vg}$ | 0.00086 | Pa$^{-1}$ |

### 3.2 Roots

Model parameters for roots (Table 3) are taken from field and laboratory data of Schwarz et al. (2010a, 2012b, 2013, 2015) for *Picea abies* (Norway spruce). Figure 4 shows root reinforcement as a function of bond elongation (both in tension and compression) for four values of tree diameter (DBH, diameter at breast height) and for three distances ($d$) from the tree trunk

10   (0.5, 1.5, and at 2.5 m). The maximum root reinforcement in tension occurs within the first 5 cm of displacement in both tension and compression. The magnitude is about 5 times higher in tension than in compression and depends strongly on the size of



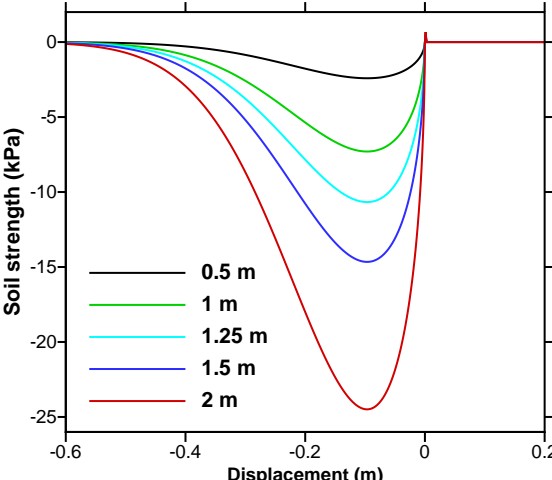

**Figure 3.** Soil strength as a function of displacement for different soil depths. Values of passive earth pressure coefficients for estimating soil compressional strength are calculated using a surface slope of $40°$. Other parameters needed for the calculation are given in Table 2. Negative values of displacement indicate compression.

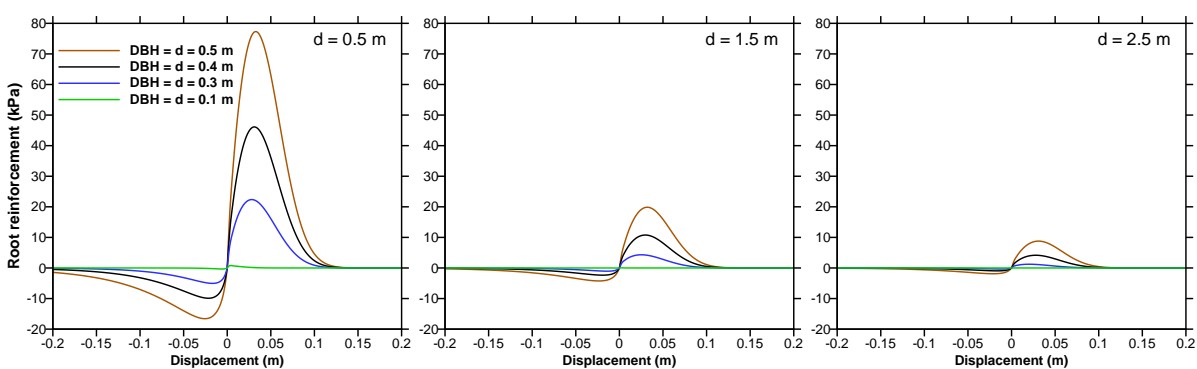

**Figure 4.** Root reinforcement as a function of bond elongation for different tree diameters (DBH) and different distances, $d$, from the tree trunk. Positive displacement indicates tension; negative compression.

the tree. Small trees (i.e., DBH $= 0.1$ m) provide negligible reinforcement at all displacements. For large trees (DBH $> 0.3$ m) lateral root reinforcement upwards of 10's of kPa is typical (Schwarz et al., 2012b). In tension, root reinforcement becomes negligible once the bond has stretched over 0.1 m, regardless of the distance from the tree trunk. In tension, the bond elongation over which reinforcement is active depends on the distance from tree and range from 0.15 m close to the tree trunk to about

5    0.05 m at 2.5 m distance from the tree trunk.





**Table 3.** Root parameters used in simulations.

| Variable | Value | Units |
|---|---|---|
| $\mu$ | 72,453 | No. roots m$^{-2}$ |
| $\eta$ | 243 | |
| $\psi$ | 18.5 | |
| $\gamma$ | -1.30 | |
| $k_0^{\mathrm{T}}$ | 356 | N m$^{-1}$ |
| $k_1^{\mathrm{T}}$ | $2.70 \times 10^5$ | |
| $k_0^{\mathrm{C}}$ | 480 | N m$^{-1}$ |
| $k_1^{\mathrm{C}}$ | $1.02 \times 10^6$ | |
| $\lambda^{\mathrm{T}}$ | 1.17 | |
| $\omega^{\mathrm{T}}$ | 2.33 | |
| $\lambda^{\mathrm{C}}$ | 1.0 | |
| $\omega^{\mathrm{C}}$ | 1.0 | |
| $\alpha^{\mathrm{T}}$ | 1.04 | |
| $F_o^{\mathrm{T}}$ | $1.5 \times 10^5$ | N |
| $\alpha^{\mathrm{C}}$ | 1.67 | |
| $F_o^{\mathrm{T}}$ | $6.5 \times 10^5$ | N |

## 4 Results and discussion

To illustrate the capabilities of SOSlope to reproduce the triggering of shallow landslides influenced by the presence of tree roots, we first present simulations of a 70 m × 70 m conceptual sigmoidal forested hillslope with a 20 m × 50 m clearing in its center. The slope is discretized into 1-m square blocks in the horizontal plane. The hillslope has a maximum slope angle of

5  40° and 32 m of vertical drop (Fig. 5a). Soil depth $D$, perpendicular to the slope surface, is 1 m and uniform across the entire slope. Trees, 50 cm in diameter (DBH), are arranged on a square lattice, 3 meters apart (horizontal distance). For the base case, the clearing has no tree and no root. Other simulations shown later include trees in the clearing. Figure 5b–d show the spatial distribution of root density for the base case for roots of three different diameters: 1, 10, and 100 mm. The hydrologic behavior of the slope, identical for all simulations, is shown in Fig. 2. Simulations are run for 2200 minutes (36.67 hours) with

10  a time step interval of 1 minute. A landslide occurs when one or more cells reach a total displacement of 4 meters. Soil and root parameters used for all simulations are those given in Tables 2 and 3.





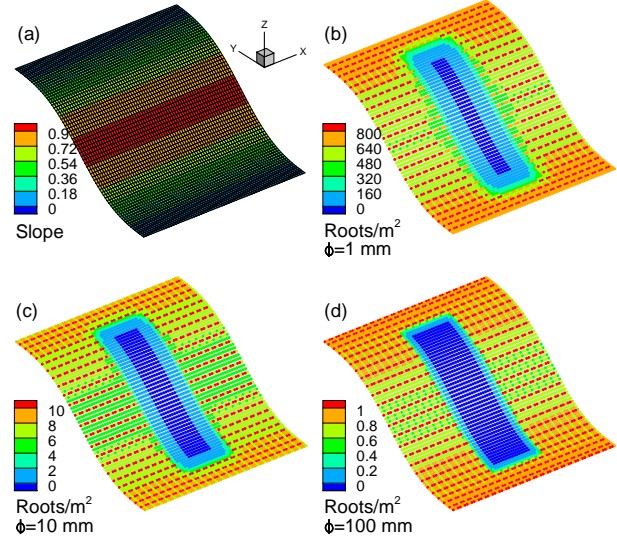

**Figure 5.** Tree-covered sigmoid slope, 70 m × 70 m, with a 20 m × 50 m clearing in its center. (a) Slope (unitless) with cell discretization (1 m). Density of roots crossing a vertical plane in units of roots m$^{-2}$ for roots of diameter (b) 1 mm, (c) 10 mm, and (d) 100 mm.

## 4.1 Displacement and force redistribution

Figure 6 illustrates the evolution of slope displacement and soil and root bond forces during loading at four different time steps, 900, 1200, 1358, and 1359 minutes after the start of loading. The last time step (1359 min) is when the slope (clearing) fails. Time step 1358 shows the slope at the time step immediately before failure. Until failure, all slope configurations are stable (factor of safety greater than 1 for all cells of the slope).

During loading, cells in the clearing move downhill more than cells in the stand (Fig. 6a–d). A discontinuity in displacement appears near the top of the clearing. This gap, 12 m long and slope parallel, occurs where the surface slope is about 0.62 (ca. 32°). This gap represents the formation of a vertical tension crack at the upper edge of a soil slip that has yet to fail completely. With increasing loading, displacement across the crack grows to exceed 1 m prior to failure (Fig. 6c). Although this crack is in the clearing in a zone devoid of trees, a few small roots from trees above the crack are present and extend across this vertical tension crack (see Fig. 5b). Cells above the crack show barely perceptible displacements ($< 0.1$ m). The situation is different in the forested area where, up to failure, displacement is significantly smaller (about 10 times smaller), uniform (no discontinuity), highest in the steepest portion of the slope (not visible in Fig. 6), with no evidence of a crack forming in the upper part of the slope. The slope in the stand remains stable after the clearing fails for the remaining of the simulation (2200 minutes). The extent of cells that have moved downhill is larger in the forested area than in the clearing: more cells uphill suffer displacement. We attribute this effect to the connected root system of trees that activates tensional forces uphill and pull rooted cells downhill. These tensional forces are absent in the clearing due to lack of roots and negligible soil tensional strength.





**Figure 6.** Time evolution of (a–d) total displacement, (e–h) downslope (parallel to steepest slope) soil force, (i–l) downslope root force, and (m–o) across-slope (lateral, also referred to as slope-parallel) root force shown at four time steps (left to right) for the slope shown in Fig. 5. Failure occurs at $t = 1359$ min (last column). $t = 1358$ min is the time step immediately preceding slope failure. Black curves in (a) indicate locations of downslope cross sections at $x = 0$, $x = -9$ m, and $x = -12$ m shown in Fig. 7.





Figure 6e–h, i–l show the downslope ($y$ axis) bond soil and root forces, respectively. During loading (Fig. 6e–g, i–k), soil compression forces increase near the bottom of the hillslope with significantly higher values in the clearing area (up to $-30$ kN in Fig. 6g, negative sign for compression). Soil tension is negligible owing to the soil minimal tensional resistance. In the forested area, roots of trees near the top of the hillslope are in tension with the tensional force increasing with increasing

loading as the slope slowly slips downhill (Fig. 6i–k). Root tension perpendicular to slope is highest on both edges of the vertical crack. This is where the largest displacements are observed generating the highest tensional forces in the roots. In that zone, tension in roots reaches almost 20 kN just before the clearing fails (Fig. 6k). Simultaneously, some roots of trees in the lower part of the slope are in compression, relieving some of the compression in the soil.

Across-slope (also referred to as lateral or slope-parallel) root forces are shown in Fig. 6m–p. Downward motion of soil in

the clearing causes a lateral tension in roots that span the transition zone from clearing to forested area. This zone is about 6–10 meters wide. It is across this boundary that displacement gradients are high and across-slope root forces highest. The lateral tension increases up to about 6 kPa with increasing downhill motion of the clearing and stays high after failure because the relative downslope displacement of cells across the slope remains.

Figure 7 yields additional insights into the dynamics and transfer of forces during loading. In that figure, values of displace-

ment (Fig. 7a–d), downslope bond force (root + soil, Fig. 7e–h), and across-slope bond force (Fig. 7i–l) are shown for three sections perpendicular to slope, at the center line ($x = 0$) that passes through the clearing, at $x = -9$ m near the left edge of the clearing, and at $x = -12$ which intersects the first row of trees next to the clearing. Figure 7a–d clearly shows the formation of the vertical crack with discontinuous displacements across it at about $y = 14$ m, initially only for the center line (black symbols), but with increasing time (or load) also at $x = -9$ m (pink symbols). The forested area ($x = -12$ m) never develops such

a crack and the displacement there is always continuous. The bond force perpendicular to slope shown in Fig. 7e–h indicates how the main resistive forces holding the slope are redistributed during loading. Initially, except for the clearing which cannot hold much tension because of lack of root, forces on the slope are in tension in the upper half and in compression in the lower part. The transition occurs halfway down the slope in the forested area (red symbols, $y \sim 0$), a little uphill at the edge of the clearing (pink symbols, $y = 5$ m). With increasing load, both tension and compression in the slope increase. Tension is highest

where root density is highest ($x = -12$) and slightly lower at the edge of the clearing ($x = -9$ m). At $t = 1200$ min, the edge of the clearing has formed a crack and forces downhill of that crack are now in compression. Roots that cross the crack at the edge of the clearing are now broken and no longer provide any tensional resistance (pink symbols in Fig. 7f).

Bonds that were in tension in the upper part of the slope at $t = 900$ min are now in compression owing to the failure of roots across the widening crack near the edges of the clearing. The clearing is now entirely held by compressive forces and by

lateral (across-slope) tensile forces shown in Fig. 7i–l. These lateral forces are due to root tensile strength and are highest near the transition from forest to clearing (pink symbols, $x = -9$ m) where the relative downslope displacement between adjacent cells is highest. Along the first row of trees (red symbols), cells that host a tree have larger values of across-slope tensional forces than cells that do not giving rise to a saw-tooth pattern of tensional force. In the clearing (black symbols), positive lateral tensional forces are entirely due to soil apparent cohesion which reaches values of almost 1 kN. With increasing load



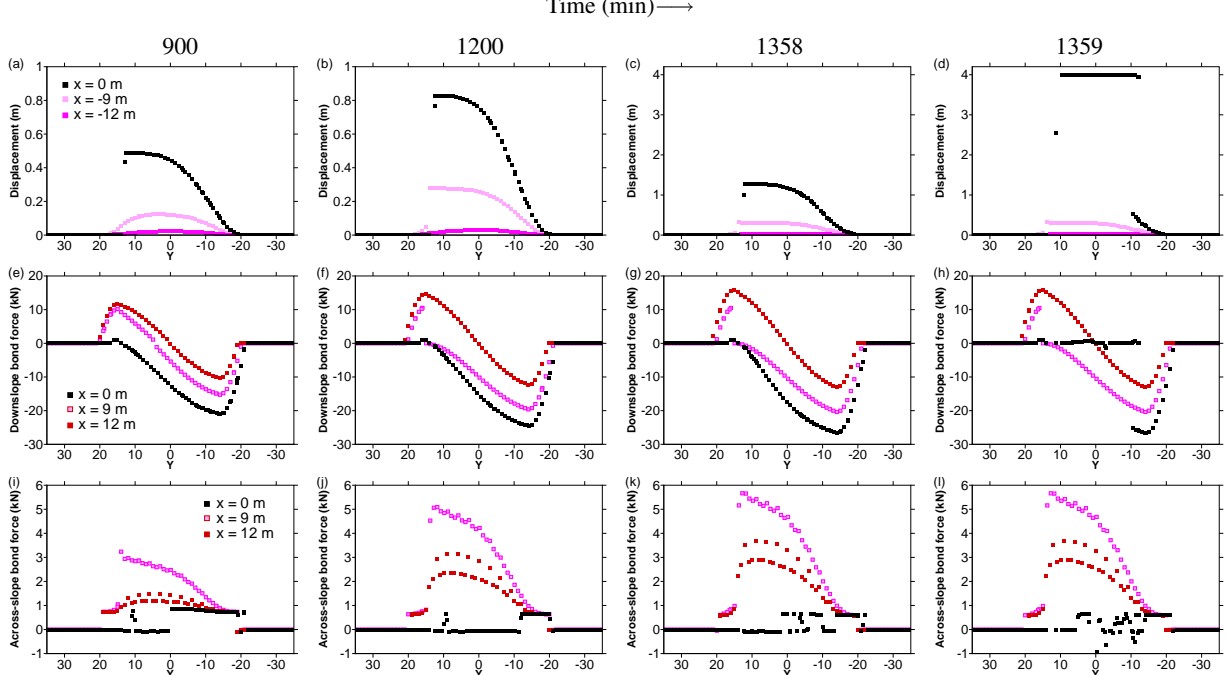

**Figure 7.** Time evolution of (a–d) displacement, (e–h) downslope bond force, and (i–l) across-slope bond force along three downslope cross sections at different distances from the center line (0, −9, and −12 meters, see Fig. 6) at four different times. Note the different scale for displacement in (c,d).

and decreasing soil shear strength due to increasing mean pore-water pressure, the clearing eventually fails at $t = 1359$ min but the forested area remains stable for the remaining of the simulation (up to 2200 min).

Results from this simulation demonstrate that maximum tensional and compressive forces in rooted slopes do not contribute simultaneously and equally to the stability of the slope during the initiation of a shallow landslide. Roots provide reinforcement

5 in tension. This tensional root force can disappears once displacement across a vertical crack becomes sufficiently large. In our example, this occurs when the crack grows to about 0.1 m (see Fig. 4). Compression is higher in the clearing (no roots) than in the vegetated area. Where present, when slope-perpendicular root tensional reinforcement is eliminated, soil stability is entirely accommodated by soil compressive resistance and by lateral tension held by roots. Lateral root forces provide additional stability to the clearing by redistributing slope-perpendicular forces laterally across the slope. The clearing fails

10 when soil strength at the base can no longer be held by the combination of the lateral root bond forces and downslope soil compression, and compression in the soil exceeds the maximum strength.

We can summarize the redistribution of forces during the loading of a rooted hillslope into three distinct phases:

1. Increasing load and weakening of soil strength along the basal failure plane (not shown) without any soil motion (factor of safety above 1).





2. Initiation of downward motion after some cells reach critical condition (factor of safety equal to 1). Force redistributions (compression in soil, tension and compression in roots) prevent the slope from failing. These forces increase with increasing load and increasing mean pore-water pressure (e.g. Fig. 7, $t = 900$ min). The culmination of slope-perpendicular tensional forces across the crack ($t = 900$ min, Fig. 7e) occurs with: (1) less-than-maximum compressive forces in the lower-half of the slope, and (2) lateral tensional forces activated at the edge of the forested area (Fig. 7i).

3. Culmination of compressive forces leading to failure when exceeded ($t = 1359$ min, Fig. 7, last column). This occurs after tensile, slope-perpendicular forces due to roots are lost across the vertical crack and when lateral root tensile forces reach their maximum values.

The timing and duration of these three phases will vary with soil mechanical properties, slope inclination, slope morphology, root distribution, and hydrology, resulting in an increase or decrease in the stability of the slope. These three phases of force redistribution are used as criteria to define the triggering of a landslide. In civil engineering, calculations using infinite slope analysis, for example, must yield a factor of safety greater than 1 for the slope to be deemed stable. Any values below one imply an unstable slope with the possibility of a landslide, even if slope motion subsequently stops with no occurrence of a runout. This definition of a landslide corresponds to the second phase of force redistribution where motion has initiated but complete failure has not yet occurred. Many such occurrences of a failed landslide (at least temporarily) exist; one is shown in Fig. 8. In risk analyses, or when studying geomorphological processes, a landslide occurs by definition only when the soil mass fails completely and is followed by a runout, corresponding to the third phase of our force redistribution process. In that case, the transition from phase 2 to phase 3 and the accompanying redistribution of forces, is the critical process.

Changes in the values of the factor of safety (FS) over time help understand the processes of landslide triggering and illustrates the three phases of landslide initiation and force redistribution. Figure 9 shows the evolution of displacement, the factor of safety, and the mean pore-water pressure with time at the center of the clearing. Initially, values of FS are larger than 1 and decrease with increasing mean pore-water pressure up until about 400 minutes. This corresponds to the phase 1 described above. Beyond 400 minutes, the value of FS remains very close to 1 and this period, which lasts until failure, corresponds to the critical state of a self-organized system before global failure (phases 2 and 3). During this critical period, the number of cell moves (not shown) increases dramatically as a results of force redistribution between bonds of connected cells and as the number of cells with factor of safety less than 1 increases. The increase of the number of redistribution with loading is similar to the process of avalanching in load-controlled self-organized systems like fiber bundle models (Cohen et al., 2009; Lehmann and Or, 2012). The increase in force redistribution across the slope corresponds to the progressive slope failure stage of coalescence of local failure surfaces that eventually leads to global failure (Petley et al., 2005; Cohen et al., 2009). This is equivalent to our phase 3.

The decrease in the factor of safety is linked to the increase in mean pore-water pressure in the soil (Fig. 9). A detailed analysis of how hydrology impacts slope stability is beyond the aim of this paper. Here we wish to point out that our simple dual-porosity model, with the coexistence of pore-water pressure in the macropores and suction stress in the matrix (see Fig. 2) is realistic and can model a wide range of hydrological situations that can lead to shallow landslide triggering. In the simulation





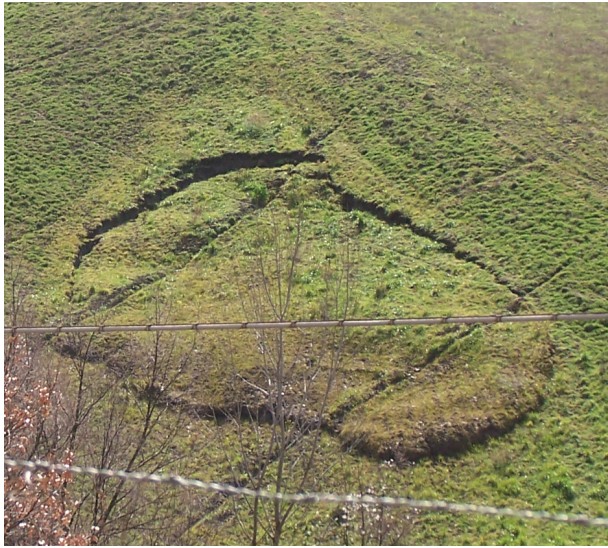

**Figure 8.** Initiation of slip at Castel Vecchio, Italy, that did not result in a landslide in the geomorphic sense, but is considered as one in the engineering sense. See text for details.

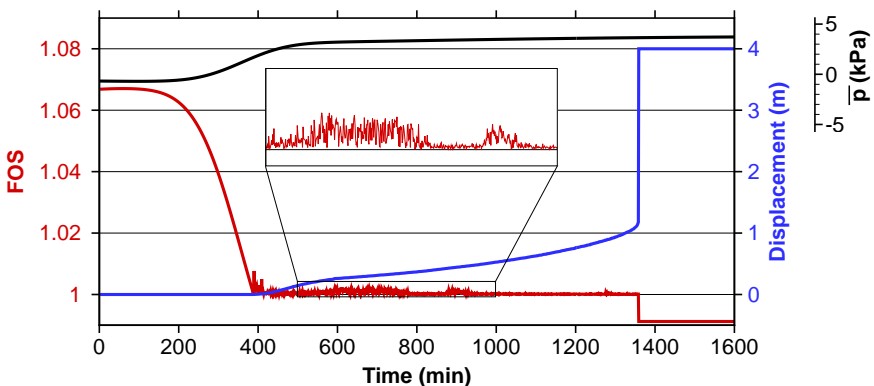

**Figure 9.** Time series of the factor of safety (red), displacement (blue), and mean pore-water pressure ($\bar{p}$, black) at the center of the clearing ($x = y = 0$) for the simulation shown in Figs. 6 and 7.

shown in Fig. 9, there is an imperceptible increase in the factor of safety during the first phase of the simulation until about 100 minutes. This increase is due to the increase (in absolute value) of the suction stress that increases the soil apparent cohesion (see Fig. 2). The increase in pore pressure after about 200 minutes causes the soil to weaken with an associated decrease in the factor of safety eventually leading to the critical state (FS close to 1). Decrease of matrix suction linked to flow of water from

5 the macropores to the matrix increases the mean pore-water pressure (Fig. 2) and eventually causes soil to weaken sufficiently



for a landslide to occur. Depending on the application of the model and on the local hydrological properties, choices of different values of hydrological parameters than those used in this example could lead to different hydrological triggering. For example, triggering could be due to the rapid increase in macropore water pressure and the saturation of the soil from top to bottom with little time for changes in matrix pore pressure to occur. In our example, preferential flow paths lead to local increases

of pore-water pressure that, in combination with a loss of suction stress in the soil matrix, resulting in a critical drop of soil shear strength typical of forested soils on compacted bedrock (Lehmann et al., 2013). Yet in another situation, high pore-water pressure can originate from ephemeral springs or water exfiltration from fractured bedrock (Montgomery and Dietrich, 1994).

## 4.2   Effects of root tensile and compressive strength

Our results show that force mobilization and redistribution in the soil and in the root system during the triggering of a shallow

landslide is a complex process. Our model can be used to investigate the effects of the various components of the bond force system (roots and soil) on the dominant reinforcement mechanisms (tension or compression, lateral or downslope) and how these forces control the stability of the slope. Understanding which forces control slope stability under certain conditions is important for making appropriate simplifications when the full level of details is not needed or not known.

Figure 10 shows the displacement for 9 hillslope simulations where trees, spaced 3 meters apart, cover the entire slope. In

Fig. 10a, tree diameter (DBH) is 50 cm; in Fig. 10b, it is 40 cm; in Fig. 10b, it is 30 cm. In each of the 3 cases, three simulations are shown: trees with roots that have both tensile and compressive strength (the standard behavior), roots that only have tensile strength, and roots with only compressive strength. All simulations are run with the same hydrologic loading used in earlier simulations (see Fig. 2).

The slope behaves differently depending on the tree size and the type of root reinforcement. Root reinforcement for the

50-cm diameter trees is sufficiently large that the slope does not fail regardless on the type of root reinforcement (tensile, compressive, or both). For the 40-cm diameter trees, there is a threshold: the tensile strength of roots is needed to keep the slope stable. Without root tensile strength (compression only), the slope fails (Fig. 10b). Finally, for the 30-cm diameter trees, all root reinforcement configurations lead to slope failure, but at different times, with compression only roots failing first and roots with both compression and tension last.

Results indicate that roots with only tensile strength limit downward slope slip under loading and delay slope failure more than roots that have only compressive strength. Roots that have both tensile and compressive strength offer the best protection against slope motion and slope failure. Neglecting root compression in the simulations results in only a couple of centimeters difference in slope displacement or less than one hour in the timing of the landslide. Neglecting root tension, however, can result in predicting a false slope failure. Also, neglecting tension misses the jump in displacement during the early initiation of

the landslide when roots across the tension gap in the upper part of the slope fail under tension (see Fig. 10c). Note also that when roots across the vertical crack fail in tension (as is the case for the 30-cm diameter trees), the slope eventually fails. This appears to be the case for all simulations we tested. However, simulations with roots that do not break across the widening crack do not necessarily remain stable over the duration of the simulation (2200 min).





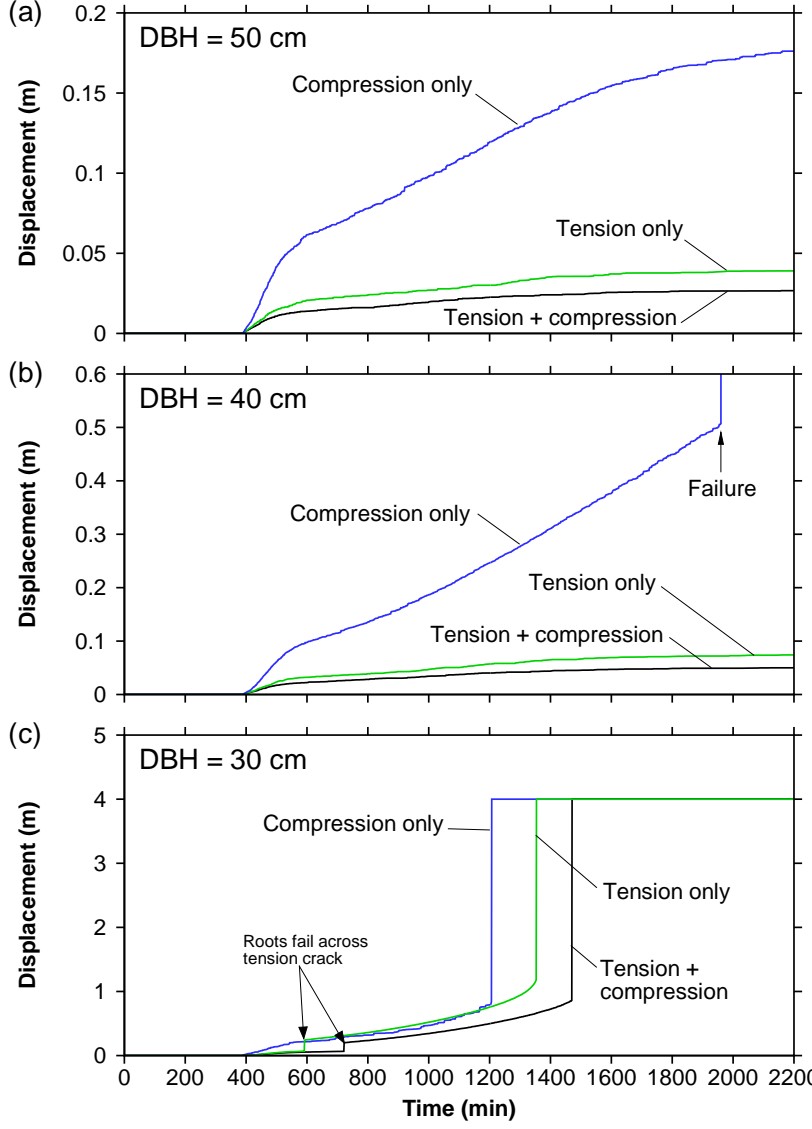

**Figure 10.** Effects of tensile and compressive strength of roots on slope displacement and stability for trees of different diameters. Displacement at the slope center ($x = y = 0$) as a function of time for (a) a stand of trees 50 cm in diameter, (b) 40 cm in diameter, and (b) 30 cm in diameter. Trees are spaced 3 meters apart in each cases. Each graph shows three curves for roots with both compressive and tensile strength, roots with only tensile strength, and roots with only compressive strength.

Figure 11 illustrates the conditions under which the slope fails for the different tree-size diameters and root-strength configurations. Each graph in Fig. 11 shows a bond force along the downslope section at the center of the slope ($x = 0$). Downslope root bond force along that section (Fig. 11a,d,g) indicates that when roots have no tensile strength (C only), roots in the lower



section of the slope bear a higher compressive load. Similarly, roots that have no compressive strength (T only) bear higher tensile loads in the upper part of the slope, but only slightly higher than roots that have both tensile and compressive strength (T + C). As expected, roots of larger trees can bear higher tensile and compressive forces owing to higher root densities and more roots of larger diameters. Downslope soil bond forces (Fig. 11b,e,h) indicate that soils in slopes covered by smaller trees

must take more of the compressive force caused by the slope downhill motion. For the 30-cm diameter trees (Fig. 11h), the soil compressive force eventually reaches its maximum value and the slope fails, regardless of the root configuration because roots hold only a small fraction of the tensile or compressive resistance that helps maintain the slope stable: roots are too few and too small for this tree size. This is also the case for the 40-cm diameter trees when roots have only compressive strength (Fig. 11d,e). Because roots do not hold any tension in the upper part of the slope, and root compression is insufficient to sup-

port much load, soil bond in compression eventually reaches a maximum and the slope fails at $t = 1960$ minutes. Figure 11c,f,i shows the lateral root force across the slope. This force is one order of magnitude smaller than the downslope force and has only a limited role in the slope stability for the cases shown here. These simulations indicate that downslope root and soil forces control slope stability which is regulated by the maximum soil compression. Roots can reduce soil compression by taking up some of the force in tension in the upper part of the slope, preventing or delaying failure. Root compression alone is insufficient

to offset soil compression in the lower part of the slope.

### 4.3   Effects of weak zones

The structure of the stand (dimension, density, and relative position of trees) plays an important role on root reinforcement and slope stability. Moos et al. (2016) found that susceptibility to landslide was higher in plots with longer downslope gaps in the tree stand and in locations where the distance to nearby trees was higher. Inversely, Moos et al. (2016) also found that

susceptibility to landslide was smaller where root reinforcement, based on tree diameter and distance from tree, was high. Weak zones, zones with low values of root reinforcement, can serve as initiation points for slope movement and control the location and size of a landslide (Schwarz et al., 2010b, 2012a). An example of a weak zone where a soil slip initiated is shown in Fig. 12. Roots around a tree provide sufficient stiffness to make the soil around the tree behave as a rigid body. The zone in between the tree and its neighbors does not provide sufficient root reinforcement and a gap opens as a result of loading (here

rainfall). Here, we explore independently how tree size (diameter) and tree spacing can affect landslide initiation and hillslope stability.

### 4.3.1   Tree diameter

Our base scenario is the simulation presented earlier with trees 50 cm in diameter spaced 3 meters apart on a sigmoid hillslope with a $20 \times 50$ m$^2$ clearing in the center. Five other simulations were run with the clearing area planted with trees of diameter

10, 20, 30, 40, and 50 cm, all spaced 3 meters apart as in the forested area surrounding the clearing. These 6 simulations are referred to as 50/0, 50/10, 50/20, 50/30, 50/40, and 50/50, where the first and second number indicate the stand tree diameters and the tree clearing diameters, respectively. Figure 13 shows the computed factor of safety and displacement at the center of the slope for these 6 simulations. The 50/10 simulation fails earlier ($t = 1266$ min) than the 50/0 simulation ($t = 1359$ min).





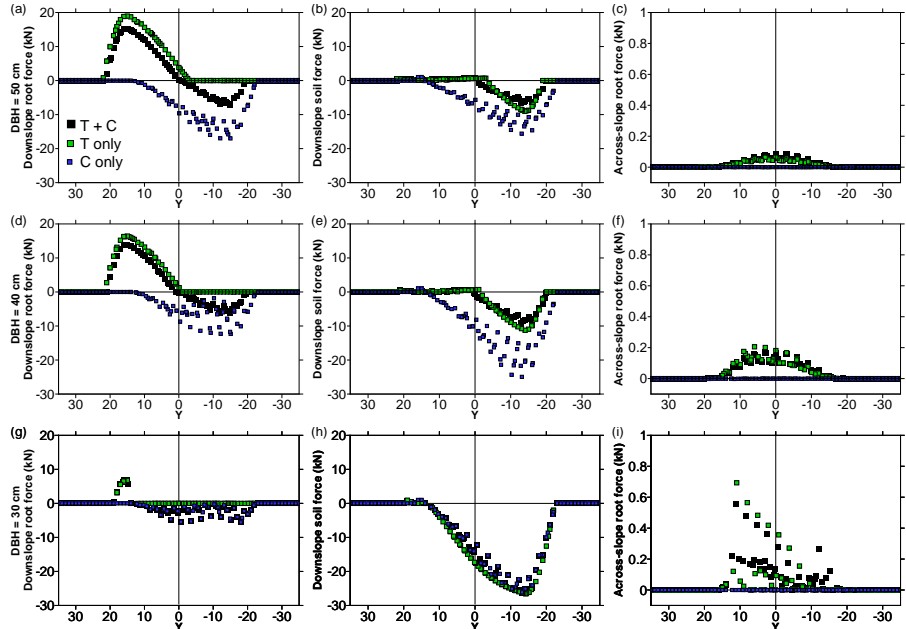

**Figure 11.** Effects of tensile (T) and compressive (C) strength of roots on root and soil bond force distribution along a downslope section at the centerline ($x = 0$) along a row of tree for a tree-covered slope with (a–c) DBH = 50 cm, (d–f) DBH = 40 cm, (g–i) DBH = 30 cm. (a,d,g) Downslope root force. (b,e,h) Downslope soil force. (c,f,i) Across-slope root force. Sections are shown either at the end of the run for simulations that did not fail, or at the time step just prior to failure for those that did (see Fig. 10).

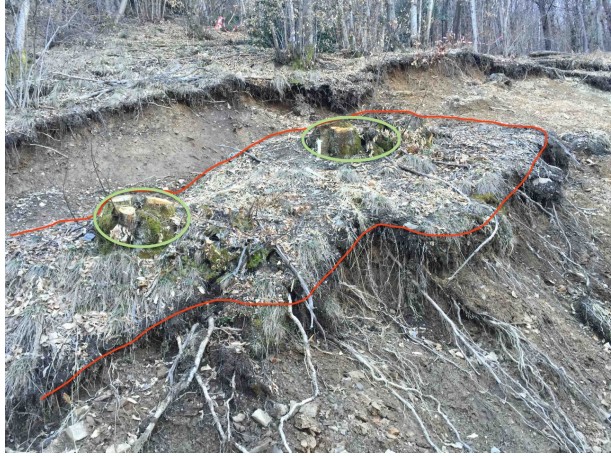

**Figure 12.** Example of a weak zone in a forested area showing isolated tree stumps with root system that behaved as a stiff island during the opening of a gap in a weak zone in between root systems of adjacent trees.




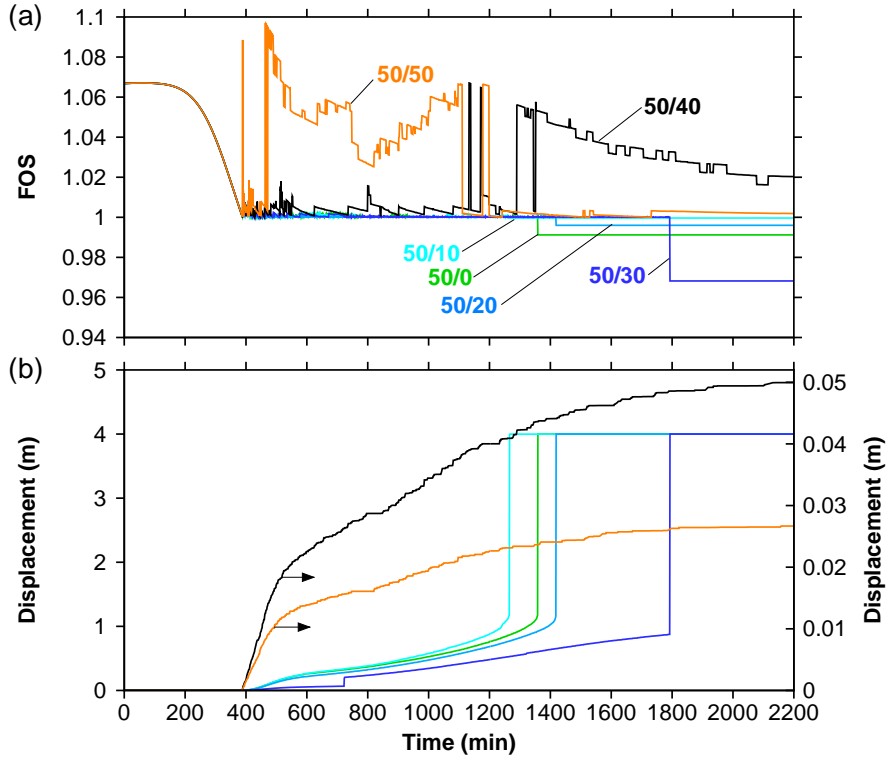

**Figure 13.** Time series of (a) factor of safety (FS) and (b) displacement at the center of the slope ($x = 0$, $y = 0$) for 6 simulations with different tree size inside the clearing. The sets of two numbers shown in (a) indicate the stand DBH and the clearing DBH in centimeters. For example, 50/10 means a stand of trees 50 cm in diameter with a clearing filled with trees 10 cm in diameter. Spacing is identical in the clearing and in the stand (3 meters). Color code is identical in (a) and (b). In (b) the simulations 50/40 and 50/50 do not fail and slope displacement plots on the vertical axis on the right side.

For larger trees in the clearing, time to failure increases, from 1419 min for the 50/20 to 1793 min for 50/30. Slopes with trees in the clearing greater or equal to 40 cm do not fail.

The time evolution of the factor of safety depends on the tree size inside the clearing. Simulations 50/40 and 50/50 have values of factor of safety that remain significantly higher than the remaining simulations, although their values sometimes oscillate very close to 1. Although these two configuration have undergone some downhill motion, it is limited to a few centimeters, significantly less than the other cases. These two slopes with large trees are in critical condition because their factor of safeties is nearly equal to 1 ($< 1.1$). Slope motion is limited to a small area near the center of the clearing and to very few cell moves owing to the large tensional resistance of roots that limit downslope movement.

Figure 14 shows the distribution of displacement, root and soil bond forces across the slope just before failure (or at the last time step of the run for simulations that did not fail), and displacement at failure for all 6 simulations (one simulation per row). For cases where a landslide occurred (0, 10, 20, and 30-cm), only the clearing area fails except for the 50/30 case



where the entire slope fails (see Fig. 14, last column). The clearing with the 30-cm trees pulls down the slope with the stand of 50-cm diameter trees. Lateral root forces in the clearing and across the stand/clearing transition, and downslope tensile forces in the stand are significantly higher for the 30-cm simulation than for any other simulations (see Fig. 14r,s). Despite smaller displacement before failure (Fig. 14p), 30-cm diameter tree roots mobilized more force than the simulation with smaller trees

owing to higher root density. This caused high downslope root forces at the upper edges of the clearing. Also, lateral force in the clearing are higher and extend across the full width of the clearing (Fig. 14s). As a result, unlike simulations with smaller trees inside the clearing, tensile root failure does not occur inside the clearing but outside in the stand, resulting in the collapse of the stand, pulled down by lateral forces originating from the 30-cm diameter trees in the clearing. This is a case where lateral tensile forces plays a crucial role: by extending spatially across a larger area, lateral forces of the 30-cm trees eventually pull

roots of 50-cm trees when the clearing fails. This numerical result helps explain the field observations of Rickli and Graf (2009) who found that the mean landslide area was greater for forested slopes than for non-forested slope in the same catchment. This behavior is also illustrated in Fig. 15 which shows downslope soil and root forces as well as across-slope root force along two downslope sections ($x = 0$ and $x = -9$ m) for the 6 simulations at the time step just prior failure (for simulations that resulted in a landslide, i.e., 10, 20 and 30-cm diameter trees in clearing) or at the end of the run (40 and 50-cm diameter trees). Results

in that figure clearly show that soil bond forces along the slope center (Fig. 15a) for small trees (0 to 30-cm in diameter) reach significantly higher values than for large trees (factor of 5 to 6), eventually reaching the soil maximum compressive strength just before failure. Downslope root-bond force is smaller for these smaller trees owing to smaller densities and smaller root sizes (Fig. 15b). For large trees, roots take up some of the load on the soil, reducing compressive forces in the soil downslope. The situation is nearly similar at the clearing-stand transition (Fig. 15d–f) except that the 30-cm diameter trees have the highest

downslope root bond force (Fig. 15, cyan symbols). Due to the comparatively larger displacement for these trees than for the smaller trees, yet significantly smaller than the 40 or 50-cm trees, the root-bond force is highest at this intermediate tree size. This is also observable on the across-slope root bond force which is highest for that size. That across-slope root force is nearly zero for the large trees (all the load is handled via the downslope bond forces). For the small trees, the across-slope root force is significant at the clearing-stand transition but small or close to zero at the center of the clearing. The 30-cm diameter

configuration stands out from the others in having the largest across-slope root bond force which eventually fails outside the clearing area, entraining the large trees in the stand during collapse.

### 4.3.2 Tree spacing

Effects of tree spacing on slope stability also yielded some unexpected results. Trees where spaced evenly on the slope using the center of the slope ($x = y = 0$) as the reference point for a tree. All other trees are located at equal intervals along the $x$ and

$y$ axes from this central tree. Figure 16 shows displacement at the slope center for five simulations with tree spacing of 3, 5, 7 (2 simulations), and 10 meters. Intuitively, one would expect that increasing tree spacing would decrease root reinforcement away from trees and increase the likelihood of a weak zone to fail. Results, however, show a different behavior. Slopes with trees spaced 3 or 7 meters (no offset, simply called 7-m spacing) apart were stable but the slope with tree spacing of 5 meters was not. Despite higher tree density than the 7-m spacing simulation, and thus having higher root density and root reinforcement values,







**Figure 14.** Effect of clearing tree size diameter on slope displacement and soil and root bond forces. From left to right, slope displacement ($d$), soil downslope compression ($F_{\text{soil}}^{y}$), downslope root force ($F_{\text{root}}^{y}$), and across-slope root force ($F_{\text{root}}^{x}$), at the time step just before failure and displacement at failure ($d_{\text{fail}}$) for the 6 simulations with different tree size diameters inside the clearing shown in Fig. 13. (a–e) Empty clearing (50/0), (f–j) DBH = 10 cm (50/10), (k–o) DBH = 20 cm (50/20), (p–t) DBH = 30 cm (50/30), (u–y) DBH = 40 cm (50/40), (z–ad) DBH = 50 cm (50/50). Outside the clearing, DBH = 50 cm. All trees are spaced 3 meters apart. Scale is given in the first row except when noted.



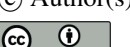

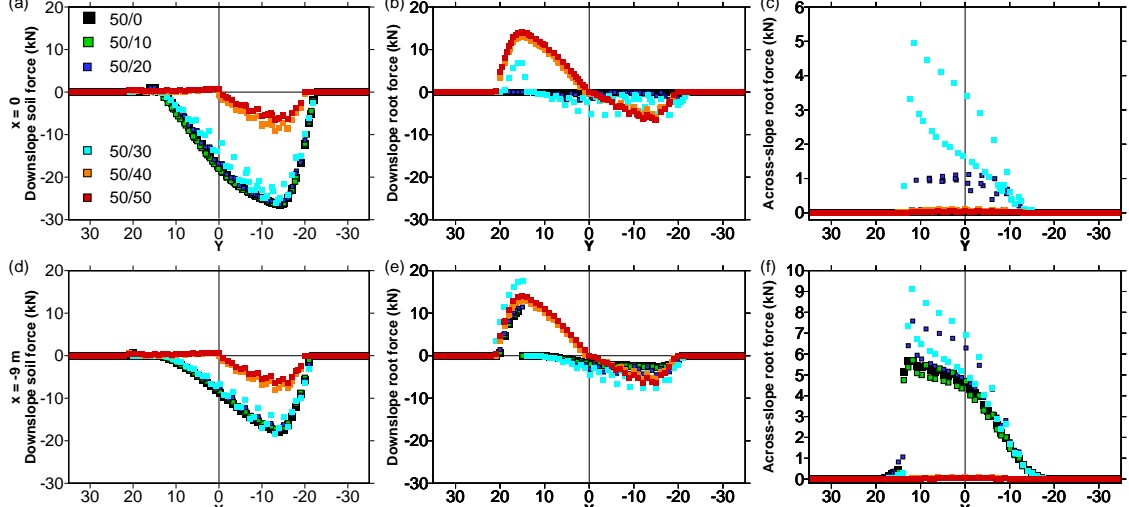

**Figure 15.** Effects of clearing tree size on (a,d) downslope soil force, (b,e) downslope root force, and (c,f) across-slope root force along two downslope sections at (a–c) $x = 0$ and (d–f) $x = -9$ m near the clearing-stand transition just before failure (simulations 50/0, 50/10, 50/20, and 50/30) and at the end of the simulations (50/40 and 50/50).

this slope failed at $t = 1781$ min while the 7-m spacing did not fail. Reasons for this seemingly strange behavior originate from the position of the tree on the slope along the $y$ axis. Because trees are spaced at regular intervals around $y = 0$, tree positions for the 5-m spacing are at $y = 0$, 5, 10 and 15 meters. For the 7-m spacing, trees are positioned at $y = 0$, 7, 14 meters. The vertical crack that forms upslope occurs at the smallest root reinforcement location in between two rows of trees. This is around

13 meters for the 5, and, 10 meter spacing, and at around 11 meters for the 7 meter spacing (the 3 m spacing had sufficiently high root reinforcement that a crack did not form). The position of the vertical crack on the slope is shown in Fig. 16, second column. The vertical crack is 2 meters higher up the slope for the 5 meter tree spacing than for the 7 m spacing. Because the crack is higher, the number of cells along the $y$ axis that move downhill due to loading is larger for the 5 m spacing than for the 7 m spacing. As a result, near the bottom of the hill, compression is significantly higher for the 5 meter spacing. With increasing

load, the 5 meter spacing slope reaches its ultimate value of compression and fails while the 7-m spacing never reaches that ultimate soil compression value. In this case, the 7 meter slope does not fail because soil compression values stabilize before reaching the ultimate soil compressive strength. This is illustrated in Fig. 17 (last column) which shows values of compression along the $y$ axis (at $x = 0$) for all the simulations.

     To explore the effect of crack location on slope stability, trees in the 7 meter spacing slope were offset 2 meters uphill, so

that a vertical crack would form at a higher elevation than without the offset. This simulation is shown with a dashed curve in Fig. 16. Figure 17d,i,n show the hillslope for this simulation and indicate that a crack forms at $y = 13$, like in the 5 meter simulation, thus resulting in high soil compression forces downslope. In this configuration, the slope eventually fails. High





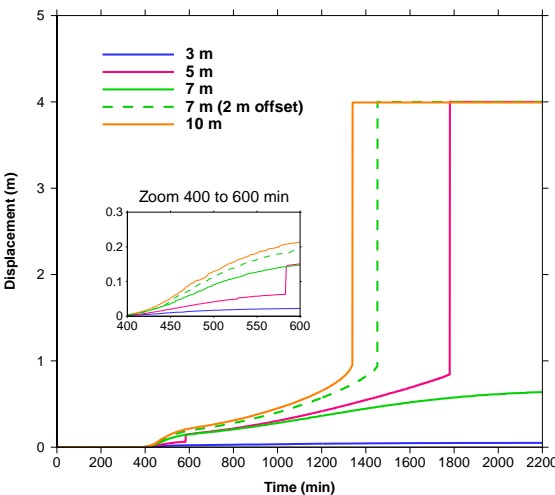

**Figure 16.** Effects of tree spacing on slope displacement at the center ($x = y = 0$) for five different tree spacings and spatial configurations.

values of soil compression forces that lead to failure (5 m, 7 m with offset, and 10 m spacing) are clearly visible in Fig 17l–o, and contrast with lower soil compression forces in simulations that did not fail (3 m, 7 m without offset, Fig. 17k,m).

### 4.4 Effects of maximum root diameter

SOSlope was used to test the influence of the range of root diameter classes on the stability of a slope. Figure 18 shows
the displacement at the center of the slope ($x = y = 0$) as a function of time for six simulations with different maximum root diameter: 5, 7, 8, 10, 20, and 100 mm. The simulations with 20 and 100 mm maximum root size diameter have no landslide and are practically indistinguishable. This is because the number of roots larger than 20 mm is insignificant and contributes little additional strength to the root bundle. The 8 and 10-mm simulations also do not fail and have only slightly larger displacements (6 to 7 cm instead of 5 cm for the 20 and 100-mm simulations). The two simulations with a maximum root diameter class of
5 and 7 mm, however, fail at 1400 and 1500 minutes, respectively. The threshold for stability is thus obtained by including root size up to 8 mm in diameter. Root reinforcement that includes only smaller roots is significantly smaller than if the entire bundle is included. Not including large roots can yield incorrect predictions of slope behavior.

Figure 19 shows the downslope and across-slope forces at the center line for several simulations with different maximum root-size diameter. The 5-mm simulation has the smallest amount of downslope root force but the highest across-slope root
force and downslope soil compression, explaining why this simulation fails while others (10, 20, and 100-mm maximum root diameter) do not. Insufficient root density and lack of large roots compromises the stability of the slope by offering little resistance to loading and declining shear strength of soil. Lateral root forces are small for all cases and has negligible impact here on slope stability (Fig. 19d).





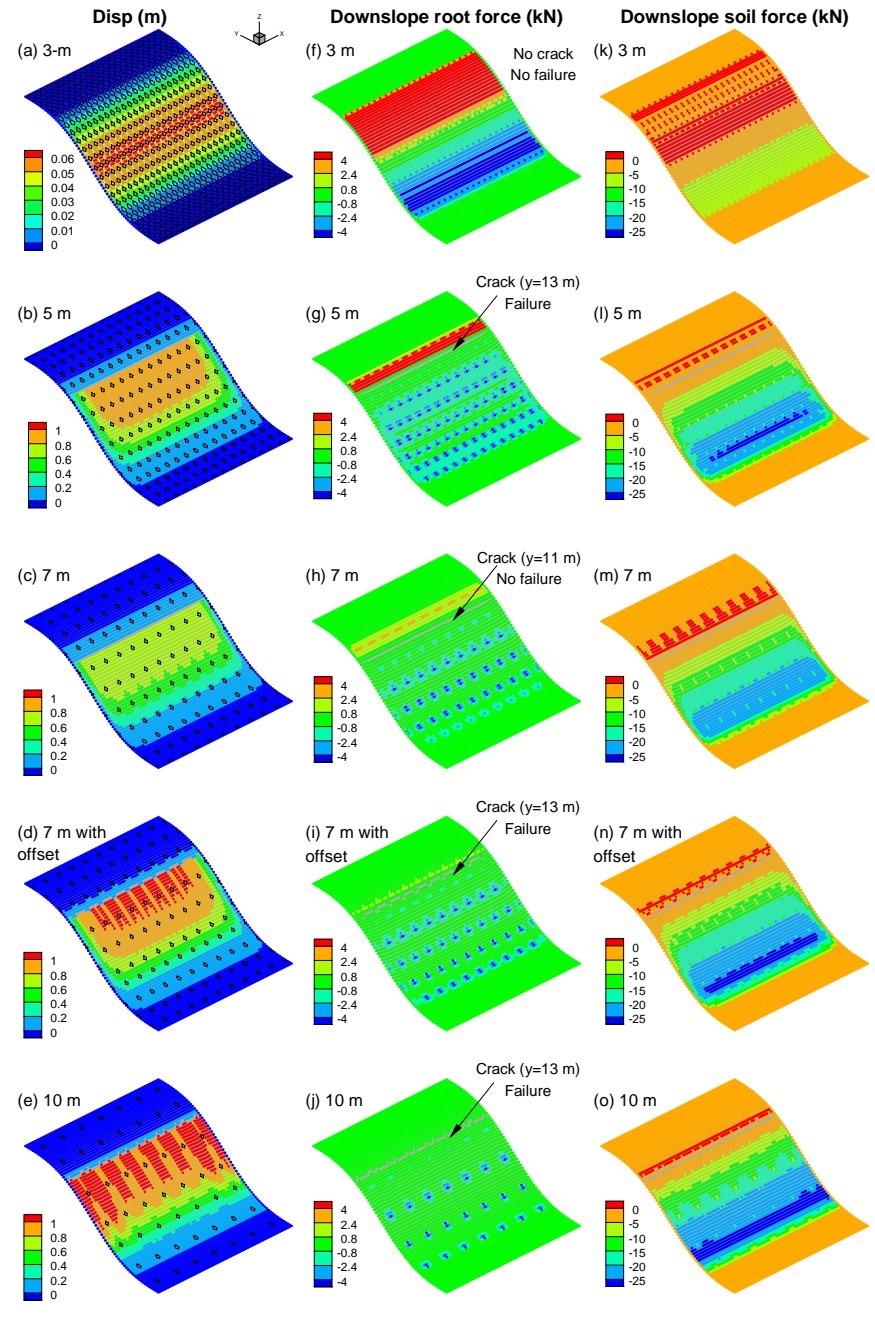

**Figure 17.** Effect of tree spacing on hillslope behavior. (a–e) Tree location (black circles) on the hillslope over slope displacement at the last stable time step or last time step for simulations where no landslide occurs (see Fig. 16). (f–j) Downslope root bond force. (k–o) Downslope soil bond force. Vertical crack position on slope shown in the the column at center.



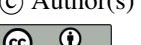

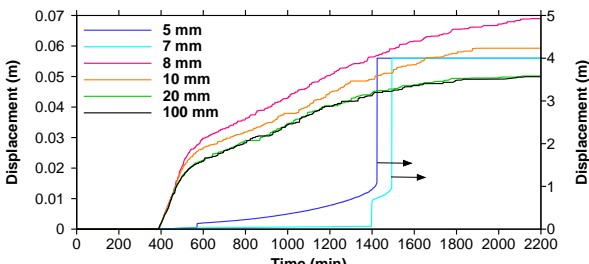

**Figure 18.** Effect of maximum root-size diameter (5 to 100 mm) on displacement at the slope center ($x = y = 0$) for a 3-m spaced, DBH = 40 cm, tree-covered slope. The 5-mm and 7-mm simulations both yield a landslide and their displacement curves plot on the vertical axis to the right.

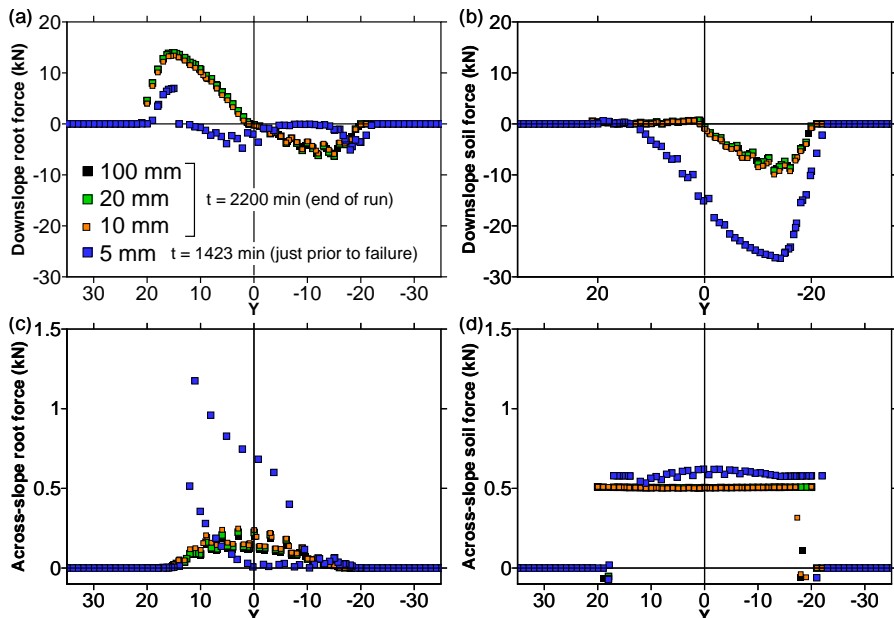

**Figure 19.** Effect of maximum root diameter on (a) downslope root force, (b) downslope soil force, (c) across-slope root force, and (d) across-slope soil force at the center line for the times specified in (a).



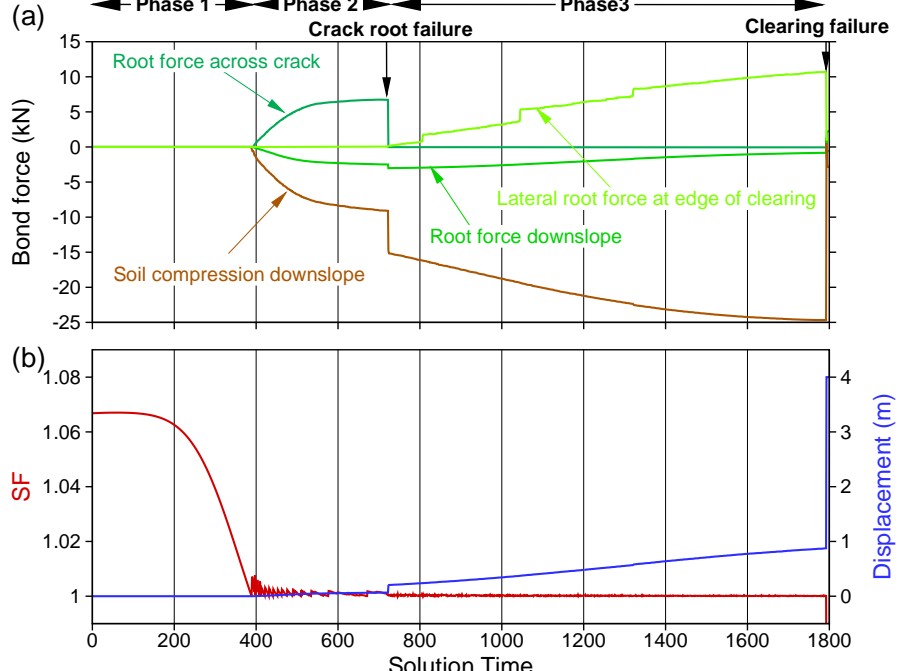

**Figure 20.** Evolution of (a) soil and root bond force, and (b) factor of safety and displacement during the initiation of a landslide in a forested hillslope (simulation 50/30 described earlier) at the center of the slope.

## 5 Synthesis of force redistributions during triggering of shallow landslides

Figure 20 summarizes the typical evolution of forces during landslide initiation of a forested slopes for the case 50/30 described in Section 4.3 (see Figs. 14–16). In this simulation, a clearing is planted with trees 30-cm in diameter while the rest of the slope has trees 50-cm in diameter.

5     The largest force that contributes to slope stability is soil compression in the area above the landslide toe. There, soil compression increases initially rapidly until it plateaus at about 700 minutes. During this increase, root tension across a growing crack increases and also plateaus. Root compression downslope similarly increases and plateau but is significantly smaller than either root tension upslope or soil compression downslope. This time period is defined as Phase 2 of our landslide initiation process which starts when many areas of the slope have a factor of safety that has decreased to 1 (Fig. 20b). Phase 1 of the

10  initiation was the decrease of the factor of safety due to loading and soil weakening without any slope motion.

    At $t = 720$ minutes, the roots across the tension crack fail and that tensional resisting force goes to zero. Instantaneously, the slope moves downhill and the force lost by tree roots is taken up by both soil and root compression downslope with the soil taking up most of the increase. This is the beginning of Phase 3. With continued loading, soil compression increases but root



compression slowly decreases. Lateral root forces at the edge of the clearing begin to take some of the load to resist downslope movement. Eventually the soil maximum compressive strength is reached and the clearing fails just before 1800 minutes.

The time span of the three phases vary with tree size, tree spacing, maximum root diameter, and of course soil and hydrological properties (here fixed for all simulations). Looking back at Figs. 13, 16, and 18, phase 2 can last from several hours
to less than one. Sometimes, no crack forms, there is no crack-root failure, and phase 2 and 3 overlap. When the slope has no clearing (as in simulations shown in Figs. 17 and 18), these same three phases exist but lateral forces play no role. Force redistribution and force balance is dominated by soil compression, adjusted by root tension in the upslope area and to a lesser extent root compression downslope. Root forces modify the force balance significantly but soil compression, due to its magnitude, dominates and controls the slope stability and its time to failure. Simulations with smaller soil depth will change this balance:
smaller small depth will decrease the absolute values of soil compression (see Fig. 3) and tree roots will then support tensile and compressive forces equal or greater to soil compression. In such a situation, roots may be the main factor controlling slope stability.

## 6  Conclusions

There are growing evidences that the effects of root reinforcement on slope stability are the results of complex interactions of
different factors in which individual contributions are difficult to isolate using classical methods (e.g. infinite slope calculations). The model SOSlope presented here is the final element of a series of related studies aiming to quantitatively upscale the stress-strain behavior of rooted soils under tension, compression, and shearing. In this framework, SOSlope represents the final module where previously investigated aspects of root reinforcements are combined to quantify the macroscopic influences of root reinforcement on slope stability considering spatial heterogeneities of root distribution. The model can produce a system-
atic analysis of the factors influencing the contribution of root reinforcement on slope stability, yielding a quantitative basis for discussion of root reinforcement mechanisms for slope stabilization and support for the assumptions or simplifications needed to implement such effects in simpler approaches for slope stability calculations (Dorren and Schwarz, 2016). Specifically, simulation results obtained with SOSlope highlight the potential of the model to investigate fundamental questions such the role of forest structure (e.g. tree size, tree spacing), root distribution, root mechanical properties, on the triggering mechanisms of
shallow landslides. Based on the results presented here the following general statements can be made:

- Maximum root reinforcement under tension and compression does not take place simultaneously;

- Root tensile strength is more effective than root compressive strength in preventing or delaying a landslide;

- The stabilization effect of roots depends on their spatial distribution: the presence of a"weak zone" leads to behavior similar to bare soils. With little or no root reinforcement, slope failure is more likely and when it occurs, occurs earlier.

- Root reinforcement at the macroscopic scale is dominated by intermediate to coarse roots when present. For the specie considered here and based on available data, roots between 5 and 20 mm contribute the most to root reinforcement.




- Tree positions in the tension zone of a potential landslide influence the stability of the slope. In general, the effect of lateral root reinforcement in tension contributes most to stability along the transition between stable and unstable zones of the hillslope where a crack can form.

To our knowledge, SOSlope is the first model to implement a new approach that characterizes the force-displacement behavior of rooted soils under both tension and compression. Including this fundamental behavior is key for understanding and modeling shallow landslide triggering. Further work is needed to extend the applicability of standard geotechnical methods (e.g. Schwarz et al., 2015) for the quantification of those soil and root forces.

The SOSlope model can be applied at the hillslope scale to investigate the effect of single factors such as root distribution and root mechanical properties (species specific) on slope stability, and quantification of bio-engineering measures and protective effects of forests. An important application at the hillslope scale is the testing of hypotheses that would support the simplification of calculations in problem-specific applications, e.g. for slope stability model at a regional scale.

The use of the SOSlope model at the catchment scale will be useful for studying the effects of vegetation on slope stability processes in the short and long terms. In the long terms, root strength can vary orders of magnitude (Vergani et al., 2016), and estimation of slope stability and landslide initiation is necessary for an integrated management of mountain catchments for risk reduction and control of sediment balance. In the short term, estimations of safety factors for rooted slopes provide important data for risks assessment in forested mountain catchments. Future work will focus on both these short and long time scales.

*Author contributions.* D. Cohen and M. Schwarz contributed equally to the model development and to writing the manuscript.

*Competing interests.* The authors declare that they have no conflict of interest.

*Acknowledgements.* We acknowledge ecorisQ members for providing on-going motivation to develop the SOSlope model and support from ecorisQ. M. Schwarz was funded by the Swiss Office for the Environment grant WoodFlow, and by the New Zealand MBIE program Growing Confidence in Forestry's Future (C04X1306).



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
