# Peer review of "Tree-roots control of shallow landslides"

_Earth Surface Dynamics, 2017_

## Referee Comment (RC1) · Anonymous Referee #1 · 20 Apr 2017

Synopsis: The paper introduces an innovative numerical model for the simulation of shallow landslides, which specifically accounts for the role of tree roots in slope stability. After a detail and comprehensive description of the model, the results of different trial simulations are presented and discussed, in order to clarify the complex interactions of forces and factors in rooted soils, then emphasizing the potential of the proposed model as a tool for reproducing the triggering of shallow landslides.

General comments: I really appreciate the content of this paper since it focuses on an extremely interesting topic (i.e. the role of tree roots in slope stability) that very few authors have dealt in detail. The manuscript is fairly well written; however, I suggest some modifications in the structure in order to improve the readability of the text. In fact, in its current form, the manuscript is probably a little bit long and in several parts a little wasteful. In detail:

[Figure]

1) The introduction is too dispersive: you should shorten this section by removing the superfluous information and focusing instead on the difficulties concerning the modelling of shallow landslides in rooted soils, then emphasizing how SOSlope is able to overcome these problems. The novelty of your model should be highlighted both in the introduction and in the conclusions of the paper;

2) In my opinion, the model description is over-detailed: considering that the main peculiarity relies on the simulation of the root-soil interaction, you should shorten the description of the hydrological part by referring more clearly to the bibliography;

3) You should clearly distinguish the "results" and "discussions" section, in order to clarify what conclusions can be drawn from the results, then avoiding undue redundancy . This modification is necessary also considering the great amount of data reported in the text. In this respect, on the basis of your conclusions you should also reconsider if all the twenty figures that you have included in the manuscript are required for the comprehension of the text.

As regards SOSlope, considering the results of your simulations it seems to be an extremely useful tool for the simulation of shallow landslides. However, there a few aspects concerning the model which I believe are well worth examining at the end of the "discussions" section. These aspects should be considered as food for thought for future improvements of the model. Specifically:

1) The slip surface is currently predefined by the user. This point should be modified in view of performing provisional analyses: are you planning to modify the model in this sense?

2) Is the model able to account for different infiltration rates by varying the vegetation type?

Specific comments:

p. 4 line 24: are you sure it's "hPa"?
p. 12 line 31: "Than 1" in place of "then 1".

p. 14 line 13. "Eq.33". Maybe it's "Eq. 31".

p. 19 line 15-16. "The extent of cells. . . . . . . suffer displacement". Please clarify this sentence.

p. 21 line 12: "kN" in place of "kPa".

p. 29 line 5: "configurations".

p. 29 line 7: "factors of safety".

p. 30 lines 20-24: These sentences are not clear. Please rewrite this part.

p. 32 lines 2-3: I think that this sentence is quite superfluous.

p. 32 line 5: Please check the commas. Which is the tree diameter used in these simulations? It is not specified in the text.

p. 32 line 6-7. Again, I think that this sentence is quite superfluous. Please try to shorten this section (4.3.2) since it is quite redundant in a few parts.

p. 33 line 14 and figure 19: why did you decide to show only the 5-mm results? What about the 7 and 8 mm?

p. 37 line 10: "smaller small". Please fix it.

p. 37 line 29: Please check the syntax of this sentence.

Table 3: F0T is repeated twice: please fix it.

Fig. 9: Which is the meaning of those fluctuations for the Safety Factor?

Fig. 13: Again: which is the meaning of those fluctuations for the Safety Factor? From the figure is not clear when the FS goes below 1. Why those two arrows are shown in the figure? The double scale on the two axes is a bit confusing. Why the failure in the 50/10 simulation occurs before the 50/0 one? The latter result is a bit odd and should

be briefly discussed in the text.

Fig. 16: I believe that the zoomed picture is quite useless, also considering that you do not discussed it in the text

Figure 18: Why those two arrows are shown in the figure? The double scale on the two axes is a bit confusing.

[Figure]

---

## Editor Comment (EC1) · R. Gloaguen (Editor) · 3 May 2017

Dear authors,

In view of the comments posted by the reviewers I would recommend that you provide some minor revisions to your submission. Provide also a clear and detailed rebuttal letter.

The key aspects that need to be attended to in priority are mainly structural:

1- Shorten and streamline the abstract 2- Focus the introduction 3- Separate results from the discussion. That is mandatory if you want a clear demarcation between facts and conjectures. 4- The discussion should include a self-assessment of the added value of this new model and a critical appraisal of the validity of the results with respect

to the model limitations.

**ESurfD**

Interactive
comment

---

## Author Comment (AC1) · 9 Jun 2017

Thank you very much for your constructive and helpful review. Below are responses to comments and how we have modified/improved our manuscript.

General comments

**1) The introduction is too dispersive: you should shorten this section by removing the superfluous information and focusing instead on the difficulties concerning the modelling of shallow landslides in rooted soils, then emphasizing how SOSlope is able to overcome these problems. The novelty of your model should be highlighted both in the introduction and in the conclusions of the paper;**

The introduction is now re-organized to separate the importance of root reinforcement for vegetated slope stability and how our model provides a new approach (Section 1, Introduction) and the motivation and background for our work and the geomorphic importance of landslide processes in general (Section 2, Background and motivation). We kept the second part because we could not find an equivalent discussion in the shallow landslide/slope stability literature. We feel this motivation and description of the geomorphic aspect of landslide and slope stability is essential and brings in the 'big picture' often needed for motivating research. We also have now better emphasized the novelty of our work in the introduction.

**2) In my opinion, the model description is over-detailed: considering that the main peculiarity relies on the simulation of the root-soil interaction, you should shorten the description of the hydrological part by referring more clearly to the bibliography**

This somewhat contradicts a comment by reviewer 2 that "the hydrologic component of the model is lacking". We have kept the details of the hydrologic model as is since parts of it are entirely new. We also feel that, since the focus of the paper is on a new model and its capabilities, a high level of details is necessary for a complete understanding of the model without the need to look into earlier literature that describes incompletely parts of our model.

**3) You should clearly distinguish the "results" and "discussions" section, in order to clarify what conclusions can be drawn from the results, then avoiding undue redundancy . This modification is necessary also considering the great amount of data reported in the text. In this respect, on the basis of your conclusions you should also reconsider if all the twenty figures that you have included in the manuscript are required for the comprehension of the text.**

In general we would agree that results should be separated from discussion. However, here, because of the quantity of results provided by our new analysis, separating results from discussion would mean going back and forth several pages for finding figures and explanatory text. We have organized our results and the associated discussion by clearly separating (using subsection headings) the various effects related to root reinforcement. We feel separating results that basically show graphics, from discussion that analyze the graphics would be more confusing, producing an unnecessary lengthening of the paper, already long. Grouping results and discussion when various aspects of a model are evaluated is commonly done (e.g., D'Odorico and Fagherazzi, 2003; Lehmann and Or, 2012) and we believe, in this specific context, that it is clearer. All figures were kept as we believe they all describe important aspects of the model results.

SOSlope comments

**1) The slip surface is currently predefined by the user. This point should be modified in view of performing provisional analyses: are you planning to modify the model in this sense?**

This is inherent to most slope stability models. This will be implemented in the model but is, at this moment, beyond the objective of this paper.

**2) Is the model able to account for different infiltration rates by varying the vegetation type?**

No but this could easily be introduced by modifying the infiltration rate as a function of an evaporation function that would depend on the type of tree/plant species, the density of roots, etc. This is an interesting effect that could be taken into account in future development of the code but that is beyond the scope of the present paper.

Specific comments

**p 4. line 24: are you sure its' hPa?** Corrected to kPa.

**p. 12 line 31: "Than 1" in place of "then 1".** Corrected

**p. 14 line 13. "Eq.33". Maybe it's "Eq. 31".** Corrected. Actually Eq. 30. Thank you for
the detailed check.

**p. 19 line 15-16. "The extent of cells ... suffer displacement". Please clarify this sentence.**
Sentence was rephrased.

**p. 21 line 12: "kN" in place of "kPa".** Corrected

**p. 29 line 5: "configurations".** Corrected

**p. 29 line 7: "factors of safety".** Corrected

**p. 30 lines 20-24: These sentences are not clear. Please rewrite this part.** Sentences
rewritten.

**p. 32 lines 2-3: I think that this sentence is quite superfluous.** Removed.

**p. 32 line 5: Please check the commas. Tree diameter used.** Modified.      Diameter  used
should now be clear

**p. 32 line 6-7. Again, sentence is quite superfluous. Shorten 4.3.2.** Removed and section
now slightly condensed.

**p. 33 line 14 and figure 19: why only the 5-mm results?** The 5-mm result shows the ex-
treme behavior of the small roots (5, 7, and 8 mm). 7 and 8-mm roots do not add
any other insight and were not included to keep the figure readable.

**p. 37 line 10: "smaller small". Please fix it.** Corrected

**p. 37 line 29: Please check the syntax of this sentence.** Modified

**Table 3: F0T is repeated twice: please fix it.** Changed to $F_o^C$

**Fig. 9: Which is the meaning of those fluctuations for the Safety Factor?** Details and reference added in main text (related to Self-Organized-Critical (SOC) oscillations).

**Fig. 13: Again: which is the meaning of those fluctuations for the Safety Factor?** See above.

**Fig. 13: From the figure it is not clear when the FS goes below 1.** When FS goes to one is either visible from the curve (a jump to a value significantly less than 1) or by a line pointing to it (e.g., 50/10 case).

**Fig. 13: Why those two arrows are shown in the figure?** Arrows refer to the vertical axes used for these curves (on right side). Now indicated in the figure caption.

**Fig. 13: The double scale on the two axes is a bit confusing.** Because of the large scale difference between the different simulations two scales for the $y$ axis were used. Without two scales, the two curves would basically overlap with the $x$ axis.

**Fig. 13: Why the failure in the 50/10 simulation occurs before the 50/0 one?** An explanation is now included in the text. This is actually an essential point that emphasize the importance of root elasticity.

**Fig. 16: Zoomed picture is useless.** Kept as is since no space is wasted. Text in figure caption added to explain the inset significance.

**Figure 18: Why arrows are in the figure? Double scale confusing.** Because of the large scale difference between the different simulations two scales for the $y$ axis were used. Meaning of arrow now added in figure caption.

**References**

- D'Odorico, P., and S. Fagherazzi (2003), A probabilistic model of rainfall-triggered shallow landslides in hollows: A long-term analysis, Water Resour. Res., 39(9), 1262, doi:10.1029/2002WR001595.

- Lehmann, P., and D. Or (2012), Hydromechanical triggering of landslides: From progressive local failures to mass release, Water Resour. Res., 48, W03535, doi:10.1029/2011WR010947.

---

## Author Comment (AC2) · 9 Jun 2017

Thank you very much for your constructive and helpful review. Below are responses to comments and how we have modified/improved our manuscript.

**Hydrologic component missing.** This is somewhat in contradiction with a comment from reviewer 1 who thought the model description was too detailed. We have kept the description of the model and its hydrological component as is.

**more clearly defining range of problems applicable.** Our model is applicable to any slope with roots and soil, more specifically tree roots (roots from grass can also be modeled with SOSlope but their importance to the overall slope stability issue is less). This is now stated throughout the paper.

[Figure]

**how results improve our understanding of root control on shallow landslides** Most of our results could simply not be obtained with a standard apparent cohesion approach. The approach does not consider lateral forces (whether from soil or root) that are key to understanding the triggering of shallow landslides on vegetated slopes. We show ample evidence, by showing specific and general examples using our model, of the difference in results we obtain when root forces are taken into account vs the standard apparent cohesion slope stability models. Illustrations and discussion are found in the abstract, introduction, main text, and conclusions.

**why the proposed model is more useful than a model that employs single value for apparent cohesion** The results obtained with our model cannot be obtained without an apparent cohesion approach. Sometimes results are counterintuitive (see examples now cited in conclusions and also discussed in main text). A direct comparison with an apparent cohesion model was not the objective of this paper but could be done as part of future work on a specific site, for example.

**Abstract condensed.** Done

**Intro p. 2 L. 18, Ref. for time scale.** Added

**Intro p. 3 L. 25-27, Time scale of 100,000+ yr.** Could not find reference with such long time scale.

**Intro p. 3 L. 21-27, Time scale of 100,000+ yr.** We think it is important to mention the main influence of vegetation on hydrologic and geomorphic systems in the context of slope stability. Text kept but format changed (list removed).

**Section 1.** We have now separated the original Section 1 into an "Introduction" and a "Background and motivation". The introduction is focused on the importance of vegetation for landslides. The background and motivation section is a general

discussion of the geomorphic importance of shallow landslide, a discussion we could not find elsewhere and that motivates our work and work on landslides in general.

**Page 4.** List removed. Now in traditional text format.

**Page 4. L. 23. Need reference.** Reference added.

**Page 6. L. 18. Need reference.** Reference added.

**Page 6. L. 26.** Rephrased to indicate to what forces this applies.

**Page 7. L. 7-9. Vertical variation in root density.** Considering that no roots are present at the considered soil depth in the presented simulations, basal root reinforcement was assumed to be zero. However, our plan is to implement the effects of vertical distribution of root reinforcement in future versions of the model.

**Page 21 L. 1-9. Define loading term.** Now defined at first mention of word.

**Page 23 L. 31-34.** We agree with the reviewer that initial soil movement can have a large effect on soil porosity and pore water pressure, but this is well beyond the scope of this work. Our focus is on roots first. Such effects could be dealt with in a subsequent version of the model, dedicated more to soil hydrology.

**Page 25 L. 1-7.** Again this is out of the scope of the paper. Our simple conceptual hydrologic model is used here only to reproduce a typical behavior of water content and pore pressure up to failure as showed in Lehmann et al. (2013). This is discussed in detailed in what is now section 3.5.

**Page 27 L. 24. Specify loading term.** Done

**Fig. 5a.** Modified. Grid removed.

**References**

- Lehmann, P., and D. Or (2012), Hydromechanical triggering of landslides: From progressive local failures to mass release, Water Resour. Res., 48, W03535, doi:10.1029/2011WR010947.

---

## Author Comment (AC3) · 16 Jun 2017

We thank you for your constructive comments on our manuscript. We have revised our manuscript taking into account comments by reviewers 1 and 2 and suggestions by the editor. Here we respond directly to the editorial comments.

1. **Shorten and streamline abstract.** Done.

2. **Focus the introduction** The introduction was re-organized to separate the importance of root reinforcement for vegetated slope stability (and how our model provides a new approach) and the motivation and background for our work (geomorphic importance of landslide processes in general). We kept the second part

because we could not find an equivalent discussion in the shallow landslide/slope stability literature. We feel this motivation and description of the geomorphic aspect of landslide and slope stability is essential and brings in the 'big picture' often needed for motivating research.

**3. Separate result from the discussion.** In general we would agree that results should be separated from discussion. However, here, because of the quantity of results provided by our new analysis, separating results from interpretation would mean going back and forth several pages for finding figures and explanatory text. We have organized our results and the associated discussion by clearly separating (using subsection headings) the various effects related to root reinforcement. We feel separating results that basically show graphics, with discussion that analyze the graphics would be more confusing, and end of lengthening the paper, already long. Grouping results and discussion when various aspects of a model are evaluated is commonly done (e.g., D'Odorico and Fagherazzi, 2003; Lehmann and Or, 2012) and we believe, in this specific context, that it is clearer.

**4. Added value of model and critical appraisal of validity.** We have added material in the main text and in the conclusion that better specifies the improvement our model brings to slope stability calculations. See details in the response to the two reviewers.

**References**

- D'Odorico, P., and S. Fagherazzi (2003), A probabilistic model of rainfall-triggered shallow landslides in hollows: A long-term analysis, Water Resour. Res., 39(9), 1262, doi:10.1029/2002WR001595.

- Lehmann, P., and D. Or (2012), Hydromechanical triggering of landslides: From

progressive local failures to mass release, Water Resour. Res., 48, W03535, doi:10.1029/2011WR010947.